



# Benchmarking and Parameter Sensitivity of Physiological and Vegetation Dynamics using the Functionally Assembled Terrestrial Ecosystem Simulator (FATES) at Barro Colorado Island, Panama

Charles D Koven[1], Ryan G Knox[1], Rosie A Fisher[2,3], Jeffrey Chambers[1,4], Bradley O Christoffersen[5], Stuart J Davies[6], Matteo Detto[7,8], Michael C Dietze[9], Boris Faybishenko[1], Jennifer Holm[1], Maoyi Huang[10], Marlies Kovenock[11], Lara M Kueppers[1,12], Gregory Lemieux[1], Elias Massoud[13], Nathan G McDowell[10], Helene C Muller-Landau[6,7], Jessica F Needham[1], Richard J Norby[14], Thomas Powell[1], Alistair Rogers[15], Shawn P Serbin[15], Jacquelyn K Shuman[2], Abigail L S Swann[11,16], Charuleka
Varadharajan[1], Anthony P Walker[14], S Joseph Wright[7], Chonggang Xu[17]

[1]Climate and Ecosystem Sciences Division, Lawrence Berkeley National Lab, Berkeley, CA, USA
[2]Climate and Global Dynamics Division, National Center for Atmospheric Research, Boulder, CO, USA
[3]Centre Européen de Recherche et de Formation Avancée en Calcul Scientifique, Toulouse, France
[4]Department of Geography, University of California, Berkeley, CA, USA
[5]Department of Biology, University of Texas, Rio Grande Valley, Edinburg, TX, USA
[6]Forest Global Earth Observatory, Smithsonian Tropical Research Institute, Washington, DC, USA
[7]Smithsonian Tropical Research Institute, Apartado 0843–03092 Balboa, Republic of Panama
[8]Department of Ecology and Evolutionary Biology, Princeton University, Princeton, NJ, USA
[9]Department of Earth and Environment, Boston University, Boston, MA, USA
[10]Atmospheric Science and Global Change Division, Pacific Northwest National Laboratory, Richland, WA, USA
[11]Department of Biology, University of Washington, Seattle, WA, USA
[12]Energy and Resources Group, University of California, Berkeley, USA
[13]Jet Propulsion Laboratory, Pasadena, CA, USA
[14]Environmental Sciences Division and Climate Change Science Institute, Oak Ridge National Laboratory, Oak Ridge, TN, USA
[15]Environmental and Climate Sciences Department, Brookhaven National Laboratory, Upton, NY, USA
[16]Department of Atmospheric Sciences, University of Washington, Seattle, WA, USA
[17]Earth and Environmental Sciences Division, Los ALamos National Laboratory, Los Alamos, NM, USA

*Correspondence to*: Charles D. Koven (cdkoven@lbl.gov)

**Abstract**. Plant functional traits determine vegetation responses to environmental variation, but variation in trait values is large, even within a single site.  Likewise, uncertainty in how these traits map to Earth system feedbacks is large. We use a
vegetation demographic model (VDM), the Functionally Assembled Terrestrial Ecosystem Simulator (FATES), to explore parameter sensitivity of model predictions, and comparison to observations, at a tropical forest site: Barro Colorado Island in Panama. We define a single 12-dimensional distribution of plant trait variation, derived primarily from observations in



Panama, and define plant functional types (PFTs) as random draws from this distribution. We compare several model ensembles, where individual ensemble members vary only in the plant traits that define PFTs, and separate ensembles differ from each other based on either model structural assumptions or non-trait, ecosystem-level parameters, which include: (a) the number of competing PFTs present in any simulation, and (b) parameters that govern disturbance and height-based light competition. While single-PFT simulations are roughy consistent with observations of productivity at BCI, increasing the number of competing PFTs strongly shifts model predictions towards higher productivity and biomass forests. Different ecosystem variables show greater sensitivity than others to the number of competing PFTs, with the predictions that are most dominated by large trees, such as biomass, being the most sensitive. Changing disturbance and height-sorting parameters, i.e. the rules of competitive trait filtering, shifts regimes of dominance or coexistence between early and late successional PFTs in the model. Increases to the extent or severity of disturbance, or to the degree of determinism in height-based light competition, all act to shift the community towards early-successional PFTs. In turn, these shifts in competitive outcomes alter predictions of ecosystem states and fluxes, with more early-successional dominated forests having lower biomass. It is thus crucial to differentiate between plant traits, which are under competitive pressure in VDMs, from those model parameters that are not, and to better understand the relationships between these two types of model parameters, to quantify sources of uncertainty in VDMs.

## 1 Introduction

Climate change-related feedbacks from the terrestrial biosphere are an important and highly uncertain component of global change (Friedlingstein et al., 2013; Gregory et al., 2009). Tropical forests may contribute substantially to these feedbacks, as vegetation dynamics within these ecosystems may lead to biome shifts and resulting changes to carbon stocks (Cox et al., 2000; Huntingford et al., 2013; Malhi et al., 2009). The majority of Earth system models (ESMs) represent vegetation through conceptual structures that are likely to inhibit realistic or accurate ecosystem responses to global change. In particular, most ESMs use prescribed vegetation distributions, and/or do not represent the functional diversity that exists within tropical forests, and/or impose static vegetation turnover times. Each of these assumptions may substantially bias model results. Prescribed biogeography does not allow models to project either the abrupt changes (Cox et al., 2000) or the long-term committed ecosystem changes (Jones et al., 2009) that may result from vegetation shifts. Conversely, assuming all tropical forests are comprised of a single set of plant traits may lead to overly abrupt changes in response to an imposed forcing, as compared to approaches that allow community-wide shifts in the trait composition of forests (Levine et al., 2016; Powell et al., 2018; Sakschewski et al., 2016). Lastly, assuming fixed turnover times for vegetation may bias the responses to both elevated $CO_2$ and climate change, as doing so does not permit changes to mortality rates that may result from changes to climate and resource competition (Friend et al., 2014; Koven et al., 2015; McDowell et al., 2018; Powell et al., 2013; Walker et al., 2015), as may be already underway in tropical forests (Brienen et al., 2015).






In addition to the above structural problems in existing ESM vegetation representations, there are enormous uncertainties due to the representation of parameters in such models (Booth et al., 2012). Typically, ESMs are run with a single set of parameters that are chosen through processes that range from formal (but limited-scope) optimization approaches to ad-hoc selection of values that give acceptable results. These parameters may or may not be measurable, and if they are measurable,

the values used in a given model may need to be scaled up and may or may not agree with observed ranges (Bonan et al., 2012; Rogers, 2014). It is crucial to benchmark ecosystem models against a wide range of observations (Collier et al., 2018; Luo et al., 2012), and at the same time to understand how sensitive model predictions are to uncertainty in the model parameters (Dietze et al., 2014; Raczka et al., 2018), so that we may better assess how much to trust a given model prediction.


Land surface models (LSMs), by virtue of their enormous scope—which typically include aspects of boundary layer turbulence, radiative transfer, soil hydrology, soil biogeochemistry, plant physiology, land management, and community ecology—have many parameters, all of which are uncertain. In this paper, which focuses on vegetation processes, we broadly separate these model parameters as belonging to two sets: those parameters that comprise a plant functional type

(PFT), which we refer to as plant traits, and those parameters that govern the environment in which PFTs exist, which we refer to as ecosystem-level parameters. The importance of this distinction is that, in a dynamic vegetation model with more than one competing PFT, while we can specify the values of the traits of each PFT, the overall trait distributions are controlled by both the trait values of the PFTs and the relative abundance of each PFT. Because the PFT abundances are themselves emergent outcomes that result from the trait values (Fisher et al., 2015), there exist complex feedbacks that

amplify or attenuate the influence of any given trait value on model predictions, as well as tradeoffs or other interactions between traits. These feedbacks greatly complicate the assessment of parameter sensitivity in the models. It is thus important to distinguish between the parameter uncertainty associated with plant traits and that associated with ecosystem-level parameters to better understand how they relate to each other and contribute in different ways to model dynamics.

This paper has three goals. The first is to describe a vegetation demographic model (VDM; (Fisher et al., 2018)) for use in ESMs, which we call the Functionally Assembled Terrestrial Ecosystem Simulator (FATES). The second is to describe FATES behavior at a testbed site at Barro Colorado Island (BCI), Panama. The third goal is to explore the sensitivity of mean-state model predictions by FATES to parameter uncertainty. Because this parameter uncertainty can show up in a number of different ways in a VDM like FATES, we are interested in trying to separate three distinct types of parametric

uncertainty: (1) the direct effects of traits on physiological predictions by the model, (2) the indirect effects of trait control on competitive outcomes, which further affect ecosystem-level processes, and (3) how non-trait parameters interact with each of these trait uncertainties to further affect model dynamics.



To do this, we first describe the model, and the data that comprise the testbed used to drive the model. This testbed includes
distributions of plant traits, most of which are based directly on observations across research sites in Panama. We then
describe a series of numerical experiments aimed at exploring the structural and parametric uncertainty in the model. These
include: (1) assessing direct control of trait uncertainty on model predictions using an ensemble of model runs with only 1
PFT per ensemble member; (2) separate ensembles where we embed FATES within two related but divergent land surface
models, the Energy Exascale Earth System Model (E3SM) Land Model (ELM) and the Community Land Model (CLM); (3)
ensembles where we add greater numbers of competing PFTs (from one to three to ten) into each ensemble member; and (4)
a set of ensembles where we compete two PFTs against each other in each ensemble member, while also varying a set of
ecosystem-level parameters that govern competition and disturbance in the model.

## 2 Methods

### 2.1 Description of the FATES Model

FATES is a size- and age-structured vegetation model, whose foundations are based on a representation of ecosystem
biophysics from the CLM4.5 (Oleson et al., 2013), a discretization of individual plant and forest disturbance dynamics based
on the Ecosystem Demography (ED) approach (Moorcroft et al., 2001), and an approach to scale from individual plants to a
forest canopy based on the Perfect Plasticity Approximation (PPA) (Purves et al., 2008); all of which were first brought
together in the CLM(ED) model (Fisher et al., 2015). Following the development of CLM4.5, FATES was created by
separating the demographic components of CLM(ED) model from CLM itself, to facilitate a more modular structure, to
combat the 'shantytown syndrome' prevalent in land surface models (Clark et al., 2017) whereby new model features are
added without a clear infrastructure for supporting the additional complexity that they bring, and to enable FATES to be used
within multiple ESMs, initially both the CLM and ELM.

The two key structural components that FATES adds to a traditional land surface model, the ED and PPA approaches, are
described elsewhere in greater detail (e.g., (Fisher et al., 2018)) so we only briefly summarize them here. ED (Moorcroft et
al., 2001) describes an approach to represent a spatially heterogeneous forest canopy comprised of individual trees existing
on a complex disturbance history by approximating the forest as a set of partial differential equations in a two-dimensional
space comprised of plant size and the age of a given location since its last disturbance event. These continuous equations are
then solved numerically by discretizing the ecosystem along each of these two dimensions: plant growth and mortality is
discretized by tracking cohorts of individual trees that have a similar size, and disturbance history is tracked as a set of
patches with shared disturbance histories; such that each patch may have several cohorts growing on it. The number of
patches and cohorts varies in time. New cohorts are generated by recruitment, existing cohort number densities are reduced
by mortality, cohorts are merged if they grow to be sufficiently similar, and cohorts are split by any processes—such as light





competition—that lead to divergence in outcomes across plants at a similar stage. New patches are generated during disturbance events by reducing the area of existing patches, and patches may be merged if their disturbance history or composition are sufficiently similar.

The PPA (Purves et al., 2008) describes an approach of organizing trees (or, equivalently, cohorts) into discrete canopy strata

by rank-ordering the trees from tallest to shortest, and defining canopy trees as those whose cumulative crown area equals that of the ground (or when combined with ED, patch area) that they occupy. (Fisher et al., 2010) added a modified form of the PPA, whereby the cohorts, rather than being strictly rank ordered in their separation between canopy and understory, were probabilistically sorted into the canopy and understory based on a function of their height.

Since the original version of CLM(ED) described in (Fisher et al., 2015), there have been numerous developments in the FATES model, which we briefly summarize here. These relate to five main areas: (1) The overall structure of the model and its modularization from the CLM; (2) changes to canopy biophysics; (3) changes to allocation and allometry; (4) changes to the representation of disturbance; and (5) changes to the canopy sorting approach. For a complete reference of the FATES model, see https://fates-docs.readthedocs.io/en/latest/index.html.


A key distinction between CLM(ED) and FATES is the modularization of the code into a separate repository, with clearly identified boundary conditions between the demography code and the rest of the LSM into which FATES is embedded. Information is passed between FATES and the LSM at two different frequencies: a biophysics frequency, with a default time step of 30 minutes, and a vegetation dynamics frequency, with a default time step of one day. Within each biophysics

timestep, the LSM provides FATES with information about the current state of the soil moisture, atmospheric radiation inputs, atmospheric thermodynamic state, and some time-averaged functions of the environment. FATES solves the photosynthesis equations for sunlit and shaded leaves, separately for each PFT, along vertical gradients both within each cohort's canopy and between cohorts residing in different canopy layers, to calculate water and carbon fluxes at the level of individual leaves. FATES then provides the LSM with transpiration, integrated canopy conductance and albedo terms,

which the LSM then uses to calculate the energy fluxes at the whole-canopy level. FATES also calculates autotrophic respiration at the biophysics timesteps, and integrates the resulting net primary productivity over the day to end up with an increment of carbon per cohort at the end of each day.

At the daily timestep, FATES sequentially allocates the daily carbon increment per cohort. If this carbon increment is

negative, the amount is subtracted from the cohort's storage pool. If the increment is positive, then the cohort allocates it, first to replenish storage, then to compensate for tissue turnover. If the remaining carbon increment is still positive, the cohort will then allocate to any organ pools that are below their allometric targets, which are intrinsic functions for a given



PFT that are defined relative to the cohort's stem diameter. If, after this, carbon still remains to be allocated, the cohort will grow its stem diameter, allocating to each pool proportionally to that pool's derivative with respect to stem diameter.


A key development since (Fisher et al., 2015) has been to modularize all allometry functions so that PFTs of different allometric functional forms and parameters can exist and compete against each other. FATES requires four distinct types of allometric models to be defined for each PFT: height, crown area, sapwood cross-sectional area, and target biomass pools. All of these are prescribed as functions of a cohort's stem diameter, which thus serves as the basic index for all allometry.

FATES currently has six separate allometric target biomass pools: leaf, stem, coarse root, fine root, seed, and storage. Of these, FATES also assumes that the target values of fine root and storage pools are both linearly proportional to the target leaf biomass pool, and that the target coarse root pool is linearly proportional to the target stem biomass pool, thus only three index target pools exist: leaf, stem, and seed. As a further simplification, FATES currently assumes that sapwood cross-sectional area at breast height is a constant fraction of a cohort's target leaf area, and thus the sapwood allometry follows the

leaf area allometry.

FATES currently allows several allometric models for determining tree height. These include: a generic power law relationship, as well as the models described in (O'Brien et al., 1995), (Poorter et al., 2006), (Chave et al., 2014), and (Martínez Cano et al., 2019). For the simulations described here, we use the (Martínez Cano et al., 2019) allometry for all

cases, which uses a Michaelis-Menten form to calculate height ($H$) from stem diameter ($D$):

$$H = \frac{aD^b}{k + D^b}$$

(1)

We use a single mean set of height allometry parameters for all PFTs in this model, with the mean based on the results from

(Martínez Cano et al., 2019): $a$=57.6, $b$=0.74, $k$=21.6.

Crown allometry ($C$) in FATES is set as a 2-parameter power law of diameter, subject to a maximum stem diameter for crown allometry:

$$C = \begin{cases} fD^g & D < D_{max} \\ fD_{max}^g & D \geq D_{max} \end{cases}$$


(2)

We treat the crown area allometry coefficient ($f$) and exponent ($g$) in the above equation as plant traits that we vary based on species-level values, as described below, and we use a single maximum size for crown allometry ($D_{max}$) for all model runs, of



cm. Plants can continue to grow past $D_{max}$, but they do so at a progressively slower rate because the GPP per individual becomes capped by the crown allometry, while the carbon cost of growth continues to increase with increasing stem size.

For the target stem biomass allometric model, FATES includes several options, including a power law of diameter, as well as the functional forms of (Saldarriaga et al., 1988), and (Chave et al., 2014), which both relate target aboveground biomass to both the stem diameter and height. For all of the experiments described below, we use the (Chave et al., 2014) aboveground biomass allometry, expressed in units of kg C per individual tree:

$$AGB = cj(\rho_w D^2 h)^p \qquad (3)$$

Where the parameters $j$ and $p$, have values of 0.0673 and 0.976, respectively, $\rho_w$ is the plant trait wood density, and $c$ is the carbon to biomass ratio in wood, which we set as 0.5 for all cases.

For target leaf biomass, we use a power law allometric model:

$$L = mD^g \qquad (4)$$

Where the leaf allometric coefficient $m$ is a plant trait that we allow to vary, as described below, and the target leaf allometric exponent $g$ has the same value as the crown area allometric exponent above. As with the crown area, we set a maximum stem diameter above which target leaf biomass remains constant, and use the same maximum diameter for both allometries. Setting the exponent on leaf biomass to be the same as that of crown area, is equivalent to asserting that a tree's (target) crown depth and leaf area index within the footprint of its crown does not vary over the course of its growth trajectory. This holds true—within a given canopy strata—even though FATES does allow SLA to vary vertically through the canopy. However, the canopy trimming logic described in (Fisher et al., 2015), as well as the relative ability of a plant to actually achieve its target leaf biomass, can lead to large differences in crown depth between the canopy and understory strata, and thus differences in crown depth can occur along growth trajectories.

For seed production, FATES uses as its target a constant fraction of NPP, once tissue turnover and storage demands have been met. This represents a biomass flux from the individual cohorts to the site-level seed pool, which then serves as a basis for recruitment flux from the seed pool to new cohorts. This fraction is a plant trait that varies among PFTs. This approach represents an extremely simplified view of reproduction, which we plan to develop further, but does at least allow us to test baseline sensitivity of the current configuration.





In early versions of FATES, the presence of understory trees that persist for long periods of time but only grow very slowly—as is observed in real forests—was difficult to achieve, because of the lack of any stabilizing term on an individual cohort's carbon dynamics. If a given cohort's NPP was even slightly negative for sufficiently long, then its storage pool

would eventually be reduced below zero, at which point the entire cohort would die. In order to prevent this and allow the model to produce multiple canopy strata, we have added a stabilizing term to the carbon budgets of trees, whereby when their storage pools become depleted, we simultaneously increase the rate of carbon starvation mortality and decrease the rate of maintenance respiration. This reduction of maintenance respiration during carbon starvation is consistent with observations of trees under acute carbon stress (Sevanto et al., 2014). Because the physiologic basis and form of this process

is poorly constrained, we use heuristic functions here to define these processes. First, we define a target carbon storage pool ($S_t$):

$$S_t = nL \tag{5}$$

Where $n$ is a parameter that linearly relates the target storage pool to the target leaf biomass $L$. If a given plant is unable to achieve its target carbon storage because of having a negative NPP at any given time, then its actual storage pool $S$ will drop below the target storage pool $S_t$, then we set both the carbon starvation mortality rate ($M_{cs}$) and the fractional rate of maintenance respiration ($R$) on the ratio of $S$ to $L$:

$$M_{cs} = \begin{cases} M_{cs,max}(1 - S/L) & S < L \\ 0 & S \geq L \end{cases} \tag{6}$$

$$R = \begin{cases} (1 - q^{S/L})/(1 - q)) & S < L \\ 1 & S \geq L \end{cases} \tag{7}$$

Where $M_{cs,max}$ is a trait that defines the maximum rate of carbon starvation mortality, and $o$ is a parameter that governs the

curvature of the respiration reduction function. Thus we implicitly assume that there is a critical storage pool $S_c=L$ that sets the total-plant storage level where mortality begins; the implied parameter $S_c/L=1$ could be made explicit, but we have left this an implicit parameter here, due to the generally weak data constraints on this at present. For the experiments described here, we use a single value, 0.01, of the $q$ parameter, and allow the maximum rate of carbon starvation mortality $M_{cs,max}$ to be a PFT trait. Because both the increase in mortality and the decrease in respiration begin when S drops below L, the

parameter $(n-1)$ thus sets the size of the carbon storage buffer that determines how much cumulative negative NPP a plant can experience before it begins to suffer carbon starvation.



In FATES, we separate as distinct traits the top-of-canopy values of maximum carboxylation at reference temperature ($V_{c,max,25,top}$), leaf carbon to nitrogen ratios (C:N), and leaf mass per area (LMA). Though these traits are highly coordinated

in plants (Wright et al., 2004), we allow this coordination to occur in FATES at the point of defining a PFT that has a specific set of trait values, rather than by imposing the trait coordination within the model itself. Exceptions to this rule include that we do define the maximum rate of electron transport at reference temperature ($J_{max,25}$) as a direct function of $V_{c,max,25}$. Also, FATES scales leaf traits vertically through the canopy so that $V_{c,max,25}$., leaf N per unit area, and LMA decrease exponentially with overlying leaf area, following (Lloyd et al., 2010) and (Kovenock, 2019). This allows shaded

leaves, which are deeper in the canopy, to be thinner and have lower maximum photosynthetic rates ($V_{c,max,25}$., Jmax) than sun-exposed, top-of-canopy leaves and maintains a fixed leaf C:N throughout the canopy, following observations (Lloyd et al., 2010).

We have generalized some aspects of canopy sorting and disturbance in FATES, as compared to their CLM(ED)

representations, where some strong assumptions were implicit in the model structure. For example, gap-phase disturbance in FATES occurs when canopy trees die. When a given canopy tree dies, or more precisely, when the rate of mortality in a canopy cohort, $m_c$, is greater than zero, the patch that previously contained the canopy trees may or may not split off newly disturbed patch area. A pair of ecosystem-level parameters, the fraction of newly dead crown area that becomes a new patch (a new parameter, $f_d$,) and the fractional understory mortality during a transition to a new patch due to disturbance, $m_{u,d}$,

control the outcomes of disturbance, as described below and in figure 1a. The rate of new patch area formation, $r_d$, equals:

$$r_d = m_c * f_d$$
(8)

When new patch area is created from an existing ('donor') patch, the new patch is initialized with a fraction of the

understory plants and litter from the donor patch. The pools from the donor patch are thus split in proportion to the fraction of the old patch area transferred to the new patch. Thus when new patch area is created, all understory cohorts in the existing patch are split, with resulting number densities in the corresponding cohorts in the new and old patches proportional to the fraction of patch area disturbed. Formerly understory trees in this newly disturbed patch may, however, be killed in the disturbance event itself, thus the $m_{u,d}$ term is applied during the disturbance event.


The $f_d$ parameter thus allows FATES to scale continuously between two endmembers in how the simulated ecosystem responds to gap-phase disturbance dynamics (fig. 1a). If $f_d$ equals 1, then the existing patch area shrinks in tandem with the reduction in tree crown area within the patch's canopy. What this means is that it is effectively not possible for trees in the understory to be 'promoted' to the canopy while remaining in a patch—their only route to the canopy is to survive that

disturbance event, whereupon they are promoted into the canopy of the new patch. We refer here to this endmember as a "pure ED" representation of disturbance (on account of it's similarity to the original Ecosystem Demography approach). At





the other extreme, if $f_d$ equals 0, then no new patch area is created and there is no horizontal heterogeneity in the system (i.e. there is only ever one patch). In this case, when canopy trees die, the entire void in the canopy created by the loss of their crown area is filled through promotion of trees from the understory within the patch. We refer to this endmember as the

"pure PPA" endmember of disturbance. Intermediate cases exist between these endmembers, where a fraction of understory trees may be promoted from within a patch while a fraction of new patch area is generated. A special intermediate case considered here, is a "bare-ground intermediate", where $m_{u,d}$ equals 1—i.e all cohorts in the understory that are transferred to a new patch are killed during the disturbance event, and thus the new patch area starts from bare ground. This bare-ground intermediate, with $m_{u,d}$=1 and $f_d$=0.5, is equivalent to the equations and PPA-type model described in (Farrior et al., 2016).

We will consider each of these three special cases—the two endmembers and the bare ground intermediate—below.

A last set of modifications since (Fisher et al., 2015) are in regards to the canopy sorting via the PPA. As described above, the original PPA (Purves et al., 2008) used a deterministic ranking of trees based on their heights, and separated them in each timestep based on whether their height was above or below the height, z*, equal to the tree whose cumulative crown area

equaled the area of the ground that trees occupied. (Fisher et al., 2010) modified this to create a probabilistic PPA whereby the relative probability of trees in a cohort (or, equivalently, the fractional number density of trees of a given cohort) being assigned to the canopy was proportional to their size raised to a parameter called the competitive exclusion parameter $c_{excl}$. In FATES, we have generalized the height sorting so that it can use either the deterministic or probabilistic sorting approach to the PPA, and discuss both versions below.

**2.2 Site-scale driving data**

All model experiments here are conducted at Barro Colorado Island, Panama. We force the model with meteorological drivers for the period 1986-2017; these data are available at https://biogeodb.stri.si.edu/physical_monitoring/research/barrocolorado. All site-level data were scanned for quality assurance and quality control (QA/QC) as described by (Faybishenko et al., 2018).


The QA/QC procedure of time-series data was performed using the R (https://www.r-project.org/) software, with the application of libraries "zoo" (Zeileis et al., 2019), "xts" (Ryan et al., 2018), "tsoutliers" (López-de-Lacalle, 2019) and "Rssa" (Korobeynikov et al., 2017). The procedure includes the following major steps—the identification of problems in the datasets (QA), and then data cleaning, flagging, and gap filling of missing data (QC). Step 1 (QA) includes an initial visual

inspection and cataloging data, determining the temporal frequency of sampling to assess data availability, and preliminary assessing data quality. Step 2 (QC) includes: processing and cleaning raw datasets; formatting timestamps; detecting and removing duplicates, bad data and outliers; gap filling of missing data; and flagging QC-ed data.



### 2.3 Plant Trait Data and Application to FATES PFT Definition

A key conceptual point in this study is that we define a PFT only as a vector of plant traits; we do not make any further a
priori assumptions about what ecological role a given PFT plays. In some of these experiments, we do diagnose properties of
a PFT that allow us to—in certain cases—make post-hoc distinctions such as "early successional" or "late successional"
PFTs, and in this manuscript all PFTs may be thought of as belonging to tropical forest tree communities, but we essentially
take a probabilistic view of PFTs here as being random draws from some continuous trait covariance matrix.  To define this
matrix, we assemble several datasets and cross-reference them based on mean values per plant species, with latin binomials
used as the reference index.

We start with two datasets describing plant traits at BCI, and at two other sites across a precipitation gradient in Panama,
Parque Nacional Metropolitano (PNM) and Fort San Lorenzo (SLZ), which are originally described in (Osnas et al., 2018;
Wright et al., 2010). Data from these sets used here include leaf lifespan, leaf mass per unit area (LMA), wood density,
mortality of 10 cm and larger trees, and leaf N content. For these datasets, we only use values for trees in the canopy stratum.
Where a given species occurs in more than one site, we use mean values across the sites.   Because these are the only
datasets that include leaf lifespan estimates, where other datasets also include an estimate of LMA for a given species, we
only use the estimates in these datasets as they will correspond to the specific individuals with which leaf lifespan is also
measured.


We add two further datasets on leaf traits, both based on canopy crane measurements at PNM and SLZ sites: (Gu et al.,
2016) and (Rogers et al., 2017; Wu et al., 2019). Each of these contain estimates of $V_{c,max}$, LMA, wood density, and leaf N
content. We use FATES temperature scaling functions to calculate $V_{c,max}$ at the reference temperature (25C) based on the
temperature at which specific $V_{c,max}$ observations were made.  Together these sets of traits describe plant variation along the
leaf and wood economic spectra, two critical axes of functional diversity (Baraloto et al., 2010; Wright et al., 2004).

Lastly, we add a dataset on crown area allometry from trees at BCI (Martínez Cano et al., 2019).  The crown area allometry
in FATES is defined with crown area, *C,* set as a power law relationship with diameter, *D*, as described above, so for each
species we use the crown area coefficient *g* and exponent *d* as reported in (Martínez Cano et al., 2019).  These crown area
traits control the overall light interception ability of plants, and how it changes over plant size, and thus are important
determinants of both baseline growth rates (for coefficient *g*) as well as the derivative of growth rates with respect to plant
size (for exponent *d*).

In total, we thus use eight traits from the observational datasets: $V_{c,max25,top}$ ($\mu mol$ $CO_2$ $m^{-2}$ $s^{-1}$), wood density (g/cm$^3$), LMA
(m$^2$ g$^{-1}$), Leaf N/area (g m$^2$), leaf lifespan (y), background tree mortality (y$^{-1}$), crown area coefficient (m$^2$ cm$^{-1}$), and crown



area intercept (unitless). We assume lognormal distributions for $V_{c,max,25,top}$, LMA, Leaf N/area, leaf lifespan, and background tree mortality, and normal distributions for wood density, crown area coefficient, and crown area intercept, with correlations between these traits as determined from the data. The full matrix of observed traits is shown in figure 2, where each dot represents a pair of mean trait values for a given species, and the histograms across the diagonal show the full
distribution of species-mean values for each trait.

In addition to the observed traits that allow us to generate prior distributions on values based on data, we also want to include parameter variation in a small set of traits that are poorly observed but that we expect to be important in model dynamics. We thus add four more unobserved trait values: the leaf biomass to stem diameter allometric coefficient (kg cm$^{-1}$), the
allometric ratio of fine root biomass to leaf biomass (unitless), the fractional allocation to reproduction (unitless), and the maximum rate of carbon starvation mortality (y$^{-1}$). For each of these, we assume no correlations with other observed traits, and we assume the first three of these are normally distributed and that the last (maximum rate of carbon starvation mortality) is lognormally distributed as we do for the background mortality trait. The choice of these additional traits are to extend the possible range of dynamics to include crown thickness, plant carbon use efficiency, understory mortality rates and
thus shade tolerance, and reproductive fecundity as possible determinants in the competitiveness of a given PFT.

We thus define a single 12x12 trait covariance matrix as the basis of all experiments described below, representing the data-constrained hypervolume from which we sample plant functional types. In all experiments, the vector of trait values that defines a PFT is sampled as a single random draw from this 12x12 trait covariance matrix. An example of resampled trait
matrix from a single model ensemble is shown in figure 3. The traits considered in this study are not meant to be comprehensive, but are meant to cover a range of processes in the model, including (a) physiology and the leaf economic spectrum, (b) allocation of biomass within a whole plant to leaves, roots, and reproduction, (c) patterns of acquisition of the primary resource, light, through crown area allometry, and (d) mortality rates in both the canopy and understory.

## 2.4 Model Testing Data

We make use of the long term forest dynamics plot census data at BCI (Hubbell et al., 1999). We use a total of five censuses here, beginning with the 1985 census. We use the census data in three ways in this paper. (1) To more rapidly equilibrate the model we initialize the forest with observed size distributions (from the 2005 census); in simulations with more than one PFT present, we use the same initial size distribution for each PFT. In order to remove the initial imprint of these initial size distributions on the model output, we integrate FATES for 200 or 300 years (depending on the experiment); after this spin-
up time the model dynamics have diverged from the initialization (e.g., fig. 4b). (2) We compare model predictions of size distributions to the census data of the forest as a whole. (3) We compare model predictions of aboveground biomass against



observations, which are also derived from the BCI census data, that are reported in (Meakem et al., 2018), which are approximately 16 (15.2-16.7) kg C / m$^2$.

We compare fluxes of gross primary productivity (GPP) as well as the sensible and latent heat fluxes to observations made with an eddy covariance system. The tower used for these measurements is 41m above ground, on a plateau on BCI. The eddy covariance system includes a sonic anemometer (CSAT3, Campbell Scientific, Logan, UT) and an open-path infrared CO2/H2O gas analyzer (LI7500, LiCOR. Lincoln, NE). Hi-frequency (10Hz) measurements were acquired by a datalogger (CR1000, Campbell Scientific) and stored on a local PC. Data were processed with a custom program using a standard

routine described in (Detto et al., 2010). GPP was derived from daytime values of NEE by adding the corresponding mean daily ecosystem respiration obtained as the intercept of the light response curve (Lasslop et al., 2010). The light curve was fitted on a 15-days moving window using a rectangular hyperbolic function (runs with friction velocity less than 0.4 m/s were excluded). Lastly, we compare LAI as predicted by FATES to observations of LAI reported in (Detto et al., 2018).

## 2.5 Model Experiment Descriptions

We define a series of model experiments here to explore parametric and structural uncertainty in the model, and how trait uncertainty can combine with vegetation dynamics to feed back on model predictions (Table 1). We first begin with single PFT experiments, randomly drawing a set of PFTs and running each of them as a separate FATES simulation. We refer to the set of such simulations, which differ only in their PFT specification, as an ensemble of simulations, and each separate FATES run as an ensemble member. The size of each of the ensembles here is 576 members, chosen as a somewhat

arbitrary number but one which balances computational costs against statistical sampling depth, while allowing one simulation per CPU on the 36-core computing nodes used for most simulations. We compare outputs from these ensembles against a set of observations (of biomass, LAI, and eddy covariance) at BCI to assess patterns of variability in the model and comparisons to observations. We perform these single-PFT ensembles, with an identical set of ensemble members each for two different model configurations: CLM-FATES and ELM-FATES, in order to further test the structural uncertainty of

embedding FATES within two closely-related, yet divergent, land models.

    We assess parameter sensitivity via direct trait control of model predictions in the 1-PFT simulations by fitting splines of each of the model predictions that we analyze as a function of each of the traits that we vary across the ensemble. We calculate the maximum potential variance explained as the fraction of variance in the predictions across the ensemble that is

predicted by the fitted spline. Because some of the traits are correlated, we also assess the minimum variance explained, which we calculate by first subtracting the variance explained by all other traits, and then assessing how much of the remaining variance is explained by the trait of interest (Xu, 2013; Xu and Gertner, 2008).





As a next experiment, we add increasing numbers of competing PFTs to the model. The premise of this is that a model can
represent plant trait diversity either through multiple realizations of the model where plants with each set of traits only
interact with plants of the same type, or alternately through allowing plants with different traits to interact with each other
through competition for resources. In a PFT-based model such as FATES, these options exist on a continuum: as we add
further PFTs to a given simulation, we increase the diversity that is resolved within each simulation, and thus, in principle,
should reduce the variability across simulations. The goal here is to ask how increasing the diversity that is resolved within
any specific simulation changes the distribution of model predictions, as compared to an ensemble approach where we only
account for diversity by non-interacting PFTs. Again, we construct each experiment as a perturbed-parameter ensemble—
where we use the random draws of parameter values to construct new parameter vectors for each model run—but instead of
including a single PFT in each ensemble member, we do 576 model runs with 3 PFTs and 576 runs with 10 PFTs, in each
case drawing all PFTs at random from the multivariate trait distributions. We refer to these as the 3-PFT ensemble and the
10-PFT ensemble, respectively.

We conduct the last (10-PFT) experiment twice. In the first instance, we force the model to maintain functional diversity by
evenly recruiting from a mixed-PFT seed pool into each PFT, thus preventing competitive exclusion and inter-generational
trait filtering. This approach still allows trait filtering to occur within the lifespan of an individual plant, but prevents any
PFT from completely excluding the others, and thus acts as a discrete-PFT analog to the continuous generation of trait
diversity approach used in the model of (Sakschewski et al., 2016). In the second instance, we allow the normal inter-
generational trait filtering to occur, i.e. each PFT reproduces only recruits of its own PFT, with no supplement so that PFTs
may go extinct.

Lastly we perform a series of 2-PFT ensembles aimed at asking whether we can identify regimes where tradeoffs in general,
and in particular early-late successional tradeoffs, lead to a degree of coexistence, after 300 years, in the model. To do this,
we again conduct 576-member ensembles where each ensemble member is comprised of PFTs that are randomly drawn from
the same trait covariance matrix. In this case, we also explore different values of the ecosystem structural parameters that
govern light competition and gap-phase disturbance dynamics, as described in figure 1 and above. The control for this set of
ensembles uses the "deterministic PPA" mode for height sorting and a "bare ground intermediate" representation of
disturbance, (which we also use in all preceding experiments). Two additional ensembles vary light competition parameters
to use probabilistic PPA height sorting with $c_{excl}$=3 and probabilistic PPA height sorting with $c_{excl}$=1. In three further
ensembles, we vary the disturbance parameters $f_d$ and $m_{u,d}$ to explore the two extreme representations of disturbance to the
"pure ED" and "pure PPA" endmembers, and in the "pure ED" case, explore the sensitivity of the model to $m_{u,d}$, or how
many understory plants are killed during a disturbance event. This parameter $m_{u,d}$ has no effect in the "pure PPA" case, since
there is no disturbance when $f_d$=0.



## 3 Results and Discussion

### 3.1 Single PFT Simulations and Comparison to Observations

A first question is how the distributions of ecosystem-level properties—such as biomass, size distributions, leaf area index
(LAI), and carbon, water, and energy fluxes—from a set of single-PFT simulations compares with observations at the site.
To answer this, we conduct an ensemble of single-PFT simulations to generate a set of possible forests, each of which is
comprised of trees sharing a single set of traits. Results from this single-PFT ensemble are shown in figures 4-6. There is a
broad range of model predictions, ranging from some ensemble members that fail to establish to others which grow to highly
productive forests.


The joint distribution of GPP and LAI (fig. 4a) shows that the overall ensemble spread is roughly centered around the
observed values (shown as ellipse in fig. 4a), though with wide spread and a tail that extends to low-productivity, low-LAI
simulations. Likewise, trajectories of biomass in these simulations (fig. 4b), where each simulation is initialized with
observed size distributions and is then integrated for 200 years to come into a quasi steady state that is determined by the
ensemble parameters, converge towards a distribution in biomass that spans the observed estimates (black line in fig. 4b).
While the ensemble distributions in LAI and GPP are roughly symmetric, albeit with a tail extending to the low-GPP, low-
LAI zone in fig. 4a, the distribution of biomass shows a tail in the other direction towards extremely high biomass forests,
with some ensemble members converging towards values that are several times of the observed.

Seasonal cycles of ecosystem fluxes, as compared to observations from the eddy covariance tower at BCI (fig. 5), show both
the wide spread of ensemble members, as discussed above, as well as two systematic model-data mismatches. The first of
these is in the shape and amplitude of the seasonal cycles: FATES simulations systematically predict a decrease in GPP
during the dry season (Feb-Apr), as compared to the eddy covariance data that do not show a systematic decrease in
productivity during the dry season. The second bias is that the FATES simulations here systematically predict a lower latent
heat flux and a higher sensible heat flux than the observations. Similar biases are also documented in (Huang et al., 2019). In
this paper, we do not try to correct these biases, which likely arise from a combination of: (a) not including a broader set of
plant traits that govern ecohydrological processes, such as those traits that govern stomatal conductance, canopy turbulence,
or rooting depth distributions; (b) not using a full plant hydraulics model (Christoffersen et al., 2016; Xu et al., 2016), and (c)
not including processes known to increase GPP during the dry seasons of tropical forests such as replacement of old leaves
with leaves with greater photosynthetic capacity (Wu et al., 2017), and/or (c) biases in the soil hydrologic modules in which
hillslope hydrologic processes are largely ignored (Fan et al., 2019). A fuller analysis of plant hydrologic traits, as well as
the structural changes to represent plant hydrodynamics and photosynthetic seasonality, are underway in FATES but beyond
the scope of this paper.



Observed tree size distributions are an emergent outcome resulting from the growth rates, death rates, and light competition parameters in a forest. In principle, the accurate prediction of diameter distributions, which follow a Weibull (approximately power function at small diameters, dropping off at larger sizes), is possible in a vegetation demographic model using the combined hypotheses of ED and PPA (Farrior et al., 2016), or through the combined ED and plant hydrodynamic hypotheses (Powell et al., 2018). The ensemble of FATES simulations shown here roughly capture the shape of the curve (fig. 6),

though again, with considerable spread and some systematic biases. The wide spread in simulations show that some trait combinations lead to outcomes with either too many or two few trees at the larger end of the tree size distribution. The more systematic bias is that most of the ensemble members show too many very large, and too few small trees, as compared to observations, suggesting an overall bias in the rates of establishment, growth, and death. The degree of this ensemble-level bias—close to an order of magnitude—shows some sensitivity to ecosystem-level parameters, as discussed further below,

which suggests modest control by representation of gap-phase disturbance dynamics and light competition parameters.

Parametric control by plant traits on several ecosystem-level model predictions is shown via variance decomposition in figure 7. While the analysis here is not meant to be as comprehensive as that of (Massoud et al., 2019), fig. 7 nonetheless shows that each of these model predictions shows sensitivity to a different set of traits, thus highlighting the complex

mapping of trait variation onto model predictions. Further, many of these model predictions show a high degree of sensitivity across several variables. To some degree, this arises because of the correlations between trait values, such as through the leaf economics spectrum, which can be seen by the large spread between the maximum potential variance explained by a given trait (closed circles) and the minimum variance explained by that trait (open circles). However, in other instances, such as for tree growth rates in the canopy, sets of relatively uncorrelated parameters, such as wood density and

the set of leaf economic spectrum traits, jointly control the rates. And in other cases, individual traits directly affect the rates predicted; an example of this is the canopy mortality rates, which in this mean-state configuration are effectively showing only the background mortality rates. Understory mortality rates are slightly more complex, with joint control of both the background mortality rates and the maximum rate of carbon starvation, as well as small contributions from the leaf and stem physiological traits. Trait control over LAI shows that, because of the combined effects of within-cohort leaf optimization,

and the potential for multiple canopy strata to exist, there is a relatively weak direct control on ecosystem-level LAI by the direct leaf to stem allometric coefficient trait; and LAI is equally constrained by the leaf economic spectrum traits that control the marginal costs and benefits of additional leaves at the bottom of the canopy, as well as a small contribution from the reproductive allocation trait, which sets how the recruitment rate and thus many small are contributing to the understory LAI.





### 3.2 Sensitivity of Results to Land Surface Model

FATES is designed to work as a modular representation of plant biophysical and community assembly processes within a host land surface model, rather than being a land surface model on its own. It has been developed out of the CLM(ED) framework described by (Fisher et al., 2015), and currently works within two related but distinct LSMs: CLM5 (Lawrence et al., in review); (Wieder et al., 2019), and ELMv1 (Golaz et al., 2019).

We repeated the ensemble described above using FATES embedded within ELMv1, and compare the ensemble predictions between the two models in figure 8. The ensembles used identical plant traits, forcing data, and other FATES parameters; however, many aspects of the LSMs differ, particularly including soil depth and the number of soil layers. Thus the two ensembles can be considered an experiment to the sensitivity of the structural representation of the physical soil environment that the vegetation are growing in. Distributions between mean GPP (fig. 8a), LAI (fig. 8b), and biomass (fig. 8c) are all similar, as are the final size distributions of the plant community (fig. 8d). This demonstrates that the diversity of plant traits used here, at least in this generally well-watered site, have a stronger control on model predictions than whatever structural divergences have accumulated in the representation of the soil environment between these models.

### 3.3 Sensitivity of Results to the Number of Competing PFTs

The above experiments each contained a single PFT in each ensemble member, and so the ensemble spread of the predictions demonstrates the global trait sensitivity of monocultural forests, in the absence of competition effects. In real tropical forest ecosystems, the enormous trait diversity exists as a mosaic of plants of different species, each competing for resources and contributing to ecosystem-level dynamics. A key goal of models such as FATES is to explore how this heterogeneity in traits at the scale of individual cohorts of plants interacts with atmospheric and soil processes to govern ecosystem fluxes and structure. Thus we want to move away from the monocultural representation to ask how trait diversity affects model predictions in the presence of competitive interactions. To do this, we conduct experiments to add greater amounts of trait diversity into each ensemble by increasing the number of PFTs in each run. We first hold disturbance and light competition parameters constant; in section 3.4 below, we vary these parameters to explore their role in governing competitive outcomes.

We calculate further ensembles, drawing plant traits from the same distribution as before, but with either three or ten PFTs per ensemble member. To separate competition during the recruitment process from competition by larger-statured plants, we first 'force' some degree of coexistence between functional types by recruiting equally into the smallest-size cohorts of all PFTs, as described above. Figure 9a-d shows a key set of model predictions for each of these simulations. For all outputs (GPP, LAI, above-ground biomass, size distributions), adding additional PFTs to each ensemble member both narrows the ensemble distribution and induces a shift towards values indicative of a higher productivity forest comprised of larger trees.



This narrowing and shifting of the ensemble distributions are separate but related outcomes of resolving trait diversity and competitive interactions. In the single-PFT case, functional diversity is only resolved across ensemble members, which are
each comprised of monoculture forests. As we add PFTs, each ensemble member better samples the observed functional diversity, so we expect that the differences between ensemble members should decrease as a result. But at the same time, competitive dynamics mean that some traits will be more competitive and therefore more strongly represented in each ensemble member. Thus the single-PFT ensemble will most evenly sample throughout the possible trait distribution, while ensembles comprised of greater numbers of interacting PFTs will unevenly sample those parts of the distribution that are
more competitive.

We can quantify these competitive effects on ensemble spread by looking at how the standard deviation of the ensemble shrinks as we add more PFTs (fig. 9e). We can formulate a null model: if competition didn't matter for a given trait, then we would expect that the narrowing of the distribution upon adding further PFTs would follow a statistical sampling relationship
for independent variables, and therefore decrease as proportional to $n^{-1/2}$, where $n$ is the number of PFTs. This null model thus represents the "selection effect" of (Tilman et al., 1997). In practice, what is observed here is a rate of narrowing with additional PFTs that is much smaller than this null model - i.e. the null model narrows much faster than the realised model outcomes. This shows that competition is an important component of the higher PFT cases, both in maintaining variability within an ensemble and in increasing the ensemble mean productivity by weighting the overall ecosystem function towards
the part of the trait distribution that is more productive.

Different variables are more strongly affected by competitive dynamics than others: of the three we show here, and comparing the 1 PFT and 10 PFT cases, the competitive effects on LAI are smaller than those for GPP, which are in turn smaller than for biomass, where increasing PFT number has very little effect on the ensemble spread. An explanation for
why the competitive effects have stronger effects on some variables than others may be the relative control of a given prediction by very competitive—and thus very large—trees. Leaf area is provided by trees of both canopy strata, and so is represented most evenly across the spectrum of the competitiveness. The relative contribution by a given PFT to GPP at the ecosystem level is roughly proportional to the fraction of the canopy that the given PFT crown occupies. Because crown area scales with diameter to the power of ~1.3 (fig 2 and (Martínez Cano et al., 2019)), and the relative proportion of trees in
the canopy to the understory will further be dominated by larger trees, GPP will be more dominated by larger trees than their relative contribution to LAI. Biomass is even further dominated by large trees: combining allometry equations 1 and 3 above implies a given plant's contribution will scale with its diameter to the power of roughy 2.1, which would imply that trees that are extremely large should more seriously impact biomass than either GPP or LAI.





The convergence of the model with increasing numbers of PFTs towards higher productivity forests than are observed demonstrates that, even with the strong assertion of neutral filtering between generations that we use in these ensembles, either the competitive filtering within each generation is still too strong or else other biases in the model which are compensated for in single-PFT simulations become evident in the more diverse simulations. This is most apparent in the tree size distributions (fig. 9d), where the 10-PFT ensembles generate many more large-statured trees than either the lower-PFT-

number simulations or the sizes that are observed.    Possible causes for this bias include: (a) that the marginal competitiveness associated with a given trait advantage in the model is too strong, as compared to more neutral dynamics that may occur in real forests (Hubbell, 2011); (b) that additional, unmeasured tradeoffs associated with the set of possible strategies—which might constrain the set of possible trait combinations to remove super-species or loser species—are insufficiently represented (Thomas Clark et al., 2018); (c) that processes which govern tree vital rates at the large end of the

size distribution are poorly represented, such as senescence strategies that are observed in forest demography (Johnson et al., 2018), (Needham et al, in review); or (d) that other density-dependent effects such as herbivory or pathogen load act to reduce the competitive success of any given species in real forests (Connell, 1971; Janzen, 1970), though such effects should be weaker for functional types than species.

We further investigate the degree of competitive filtering within and between generations by re-running the 10-PFT ensemble, but in this instance, we allow species to go extinct by re-coupling the rate of recruitment of a given PFT to the seed production by that PFT.    Comparisons of the resulting predictions (fig. 10a-d) show only subtle differences in the ecosystem-level rates investigated here: biomass and GPP are barely shifted, while the distribution of LAI is slightly expanded towards higher values, and the number of small trees is slightly decreased when we allow intergenerational

competition to play out. Thus the effects of trait filtering during recruitment are much more muted in the model than the trait filtering that happens after recruitment has occurred.  This can further be illustrated if we compare ranked abundance curves for the two ensembles of trees greater than 1cm vs trees greater than 10cm (fig. 10e-f): at 1 cm, the presence or absence of recruitment filtering leads to a marked change in the slope of ranked abundance curves, whereas at 10cm the slopes of the two cases are roughly similar. Even when we force the model to allow neutral filtering during recruitment, by the time trees

grow to 10cm, the resolved filtering is strongly evident.

### 3.4 Regimes of coexistence and their sensitivity to disturbance and light competition parameters

In order to represent shifts in plant trait distributions at a given location under global change pressures, a model like FATES must be capable of maintaining some degree of trait heterogeneity in the first instance.  The maintenance of functional diversity in ecosystems is a complex topic (Chave, 2004; Chesson, 2000), and it's analysis in the context of Earth system

type models such as FATES is in its infancy (Fisher et al., 2018).  Here we seek to first understand which combinations of traits within FATES allow stable coexistence of PFTs in the mean state, and whether there is other ecosystem-level





parametric control on these regimes of coexistence. In particular, we expect that a model that resolves heterogeneity in the light environment can accommodate at least two niches, for fast-growing early successional plants, and shade-tolerant, slow-growing plants (Moorcroft et al., 2001). We can represent such a tradeoff as a line connecting two points that represent two

sets of PFT vital rates in a growth-mortality space (fig. 11); while we expect that only the combinations that define a tradeoff—i.e. a positive slope— between growth and mortality can stably coexist, we do not know what the range of possible stable lines might be. To investigate these questions, we conducted a series of 6 sets of paired-PFT ensembles (last six rows of table 1), each using the same 576 2-PFT pairs, but with different values of ecosystem level parameters that govern light competition and disturbance.


There are many different ways that a plant can grow quickly or slowly (fig. 7). This creates a problem in trying to map sets of plant traits directly onto the potential for a given pair of trait combinations to coexist with each other. To overcome this, we first want to reduce the problem from the high-dimensional set of plant traits that we use to define a PFT, to a lower-dimensional set of PFT vital rates that may act to determine the coexistence dynamics. The simplest set of rates to propose

are growth and mortality rates of canopy trees. For each set of traits that comprise a PFT, we evaluate the mean growth and mortality rates for a tree of that PFT, conditional on the tree being approximately 20cm size and located within the canopy strata of the forest. This permits a mapping between the 12 dimensional trait space and a 2-dimensional growth vs. mortality space (fig. 7). Within this reduced space, we can evaluate the slope of lines connecting pairs of competing PFTs, as in fig. 11, to identify the range of slopes that permit coexistence between PFT pairs. An example of this is shown in fig. 12.


In fig. 12a we show the lines connecting paired PFTs for a subset of ensemble members in the reference (deterministic sorting, intermediate bare ground) case. We first identify the canopy growth and mortality rates (of 20 cm diameter trees), and examine only combinations with a positive slope in a growth-mortality space, i.e. ones where we can classify an early and late successional PFT where the early successional PFT has both higher growth and higher mortality rates than the late

successional PFT. We color the lines based on whether, after 300 years, there is a degree of coexistence (which we define as having less than 95% of the biomass in either of the PFTs), and if not, which PFT is dominant. The slope of the lines shows evident control on the competitive outcome, with high slope lines dominated by early successionals, moderate slopes having some coexistence, and low slopes dominated by late successionals.

To begin to quantitatively compare the effects of the ecosystem level parameters on these competitive outcomes, we can first plot the fraction of biomass in each ensemble member existing in an early successional PFT against the log of the slope of the line connecting the two PFTs in this growth-mortality tradeoff space (fig. 12b). The points follow a roughly sigmoidal shape, again showing that low slopes (i.e. small difference in growth, large difference in mortality) lead to a competitive exclusion by the late successional PFT, large slopes (large difference in growth, small difference in mortality) leads to

competitive exclusion by the early successional PFT, and intermediate slopes can either lead to coexistence or exclusion by


either of the PFTs. Following this pattern, we then fit a logistic function to the ensemble of growth-mortality tradeoff slopes and coexistence states.

We can then compare the effects of the different ecosystem structural parameters by comparing the resulting fitted logistic curves for each ensemble (fig. 12c). The differences between these curves indicate the tendency for a given set of ecosystem parameters to favor PFTs with traits and the resulting set of vital rates that make them either early or late successional: curves with a midpoint that is shifted to the left in fig. 12c favor early successional PFTs, and those with a midpoint shifted to the right favor late-successional PFTs. For height sorting parameters, the more probabilistic the height sorting, the more it favors late successional PFTs. This makes sense: at the margin, if growing tall more quickly than its neighbors is less likely

to make a tree end up in the canopy, then that means that outliving its neighbors becomes relatively more important. The converse is also true, in that the rapid growth of early successional trees becomes more important if even a tiny difference in growth pays off with a position in the canopy.

For disturbance parameters, the story is slightly more complicated: in the case of no gap-generating disturbance (the "pure

PPA" disturbance case), early successional strategies are highly unlikely to pay off as there is no environmental niche for those PFTs to occupy. The converse is also true for the high-disturbance "pure ED" case, which is the most conducive to early successional PFTs as long as the disturbance generates bare ground (i.e, $m_{u,d}$=1) for new recruits to exploit. But if we reduce the intensity of disturbance by allowing a fraction of trees in the understory to survive disturbance events (by setting $m_{u,d}$ to 0.5), doing so effectively counteracts the increased niche area for the fast-growing, fast-dying trees by giving slow-

growing understory trees a chance to end up in newly-created patches and dominate them. Thus the "bare-ground intermediate" ($f_d$=0.5, $m_{u,d}$=1) and "pure ED" with fractional understory mortality ($f_d$=1, $m_{u,d}$=0.5) disturbance cases are relatively similar in their relative tendency to promote success between early and late successionals.

These ecosystem-level parametric differences in the balance between competitive outcomes are large: over an order of

magnitude in growth-mortality tradeoff slopes separates the midpoint of the logistic regressions between the various cases in fig. 12c. Because parameters such as $f_d$ and $m_{u,d}$ are poorly constrained at present, they represent a significant source of uncertainty in model predictions; constraining these parameters with census data thus represents an opportunity for reducing this uncertainty. Furthermore, looking at the sensitivity of the relative success of species with different growth and mortality rates across gradients of disturbance intensity or frequency may provide further benchmarks of models of this type.


Canopy growth and mortality rates are only one possible set of plant vital rates that may determine coexistence. If, instead of using canopy growth and mortality rates as the dependent variables to explain competitive outcomes produced by FATES, we substitute canopy growth rates and understory mortality rates, as may be expected given the importance of shade



tolerance in defining successional strategies, we do not see a clear sigmoidal pattern as in fig. 12. Thus, within the FATES
predictions explored here, canopy mortality rates are more determinative of success than understory mortality rates.

Overall distributions of ecosystem-level model predictions (fig. 13) are relatively similar to the earlier experiments, though
some differences can be seen. GPP distributions are similar between the cases. LAI distributions are slightly shifted
towards higher values in the probabilistic height sorting relative to the deterministic height sorting cases, and are lower in the
"pure PPA" disturbance case, likely because of overall suppression of the understory in the absence of disturbance. Biomass
distributions are shifted towards lower values in the probabilistic height sorting cases, as well as in the "pure ED" case with
$m_{u,d}$=1, and to higher values in the "pure-PPA" disturbance case. The height sorting appears to have little effect on size
distributions, while the disturbance parameters have a strong effect: the "pure-PPA" disturbance case has a greater deficit of
small trees, while the "pure-ED" disturbance case has greater number of trees in the smaller size classes (but still not as
many as observed). These effects on size distributions make sense from the perspective of small trees in each of these cases.
In the "pure-PPA" disturbance case, no new patches are created, and so there are no gaps in which small trees can grow. In
the "pure-ED" disturbance case, when the $m_{u,d}$—the parameter that controls the fraction of small understory trees that both
survive the death of a canopy tree above them and find themselves in a newly-opened patch—is 0.5 (thus representing a
medium intensity to disturbance), it provides an additional pathway for plants that are recruited into older patches to make it
to the canopy. In the higher-intensity bare-ground ($m_{u,d}$=1) and "pure-PPA" disturbance cases, the only such pathway for
plants recruited into older patches is for them to persist in the understory and grow through to the canopy, which fewer of
them are able to do.

The difference in forest structure that results from these ecosystem-level parameters can be further seen in fig. 14, which
shows in a quantitative way the ecosystem structure as sketched out in fig. 1 for a single ensemble member of each of the
cases in figure 12, which maintained some degree of early-late PFT coexistence in each of the different cases. In each of fig.
13a-f, the FATES patches and cohorts are drawn out, rank ordered by height with the tallest to the right within each patch,
with cohort width proportional to the crown area occupied by each cohort, and with patches similarly arrayed with oldest to
the right and the patch width proportional to the patch area. Thus the width of all canopy cohorts in a closed-canopy patch
equals the width of the patch that they occupy. Cohorts are colored by PFT (color) and canopy position (shading), with
yellow-green representing an early successional PFT and blue-green representing a late-successional PFT, and darker
shading of each for the understory cohorts. Shown are the final year of a 600-year set of simulations, started from bare
ground initial conditions. Differences between the cases in evident in the resulting structure of the forests. The fractional
coverage of PFTs roughly follows the pattern in fig. 12. The relative heterogeneity of patch area follows the $f_d$ parameter,
with most heterogeneous patches when $f_d$=1 and no heterogeneity when $f_d$=0. Reducing the disturbance intensity parameter
$m_{u,d}$ from 1 to 0.5 causes a small number of large trees, which had been in the understory prior to disturbance, to remain even
in newly-disturbed patches, thus making the character of patches more similar across ages. Shifting the height sorting to a

more probabilistic treatment shifts the relative size distributions of canopy and understory trees within any patch.
Animations of annual snapshots of one of these ensemble members is in supplementary videos (SV1), which shows the

emergence of heterogeneity in structure and composition over time. Figures 12-14 demonstrate the wide range of outcomes,
both in terms of PFT composition and in the size and age structure of the forest, that result from these ecosystem-level height
sorting and disturbance parameters.

**4 Conclusions**

Land surface and ecosystem carbon models are highly dependent on parameters that are both imperfectly known and that

may have highly diverse values within any given ecosystem. We attempt to separate some of these different controls on
model dynamics by distinguishing plant trait variation from other ecosystem parameters, to explore how representing
diversity in plant traits affects predictions made by a VDM, and how ecosystem level parameters govern competitive
outcomes and other predictions by the model.

In a single-PFT configuration, where competitive pressures on trait values are not present, the model exhibits both some
agreement and some biases as compared to a set of observations that span from physiological processes to ecosystem
structure. The degree of agreement with observations is not sensitive to the choice of two related land models in which we
run FATES, which both behave similarly.

As we add the effects of competitive pressures on parameter uncertainty, by increasing the number of PFTs competing
within any given simulation, these shift the distributions of model predictions in a systematic way. Productivity and biomass
increase as we add further PFTs to a simulation, in ways that push the model, which agrees roughly with observations of
biomass and productivity in a single-PFT configuration, further from the observations as we add more diversity, even though
such increased diversity in the model should better represent processes that exist in species-rich tropical forests. This

emphasizes the need to better represent tradeoffs that equalize competitive performance among species, so as to limit the
competitive ability of any given functional type to outcompete other types. These effects of competition are only partially
dependent on filtering that may occur from one generation to the next, as they are also strongly present even when we
prevent advantages in population numbers to be passed on from one generation to the next.

We further explore the effects of non-trait parameter variability on competitive outcomes in a set of paired-PFT experiments
to show how the competitively successful strategy between early and late successional traits shifts as a result of ecosystem-
level parameters. In particular, the parameters that govern both disturbance and competitiveness for light have strong effects
on the balance between early and late successional PFTs: increases to either the extent or severity of disturbance, or to the
efficiency of height-based light competition, all act to shift the community towards early-successional PFTs. These

differences in the PFT composition of the modeled forests feeds back onto ecosystem-level predictions of states and fluxes by the model.

In order to understand how global change pressures will affect ecosystems, and in turn how ecosystem response will further feed back on global change, we must consider the role of shifts in community structure. VDMs are a promising tool to
resolve these processes, however the results shown here underscore the need to better understand the roles that uncertainty in model parameters play -- both the direct role, as well as the indirect roles that govern how parameter uncertainty changes competitive pressures on trait distributions at the ecosystem level. It is thus crucial to understand and integrate these types of uncertainty into projections of the Earth system.

**Acknowledgements**

This research was supported as part of the Next Generation Ecosystem Experiments-Tropics, funded by the U.S. Department of Energy, Office of Science, Office of Biological and Environmental Research. CK also acknowledges support from the DOE Early Career Research Program. LBNL is managed and operated by the Regents of the University of California under prime contract number DE-AC02-05CH11231. AR and SPS were also supported through the United States Department of Energy contract No. DE-SC0012704 to Brookhaven National Laboratory. MCD was supported by NSF 1458021. Oak Ridge
National Laboratory is operated by UT-Battelle, LLC, under contract DE-AC05-00OR22725 with the US Department of Energy. Pacific Northwest National Laboratory is operated by DOE by the Battelle Memorial Institute under contract DE-AC05-76RL01830. MD was supported by the Carbon Mitigation Initiative at Princeton University.

**Code Availability**

The FATES model is available at https://github.com/NGEET/fates. Experiments here are based off of git commit 0bc7a5d on
the fork https://github.com/ckoven/fates. FATES is run here within two host land surface models, CLM5 and ELMv1, available at https://github.com/ESCOMP/ctsm (git commit b9c92b7) and https://github.com/E3SM-Project/E3SM (git commit 544db3b), respectively. Scripts to initialize parameter files and analyze model output shown here are available at https://github.com/NGEET/testbeds, and scripts to run the all model experiments here are available at https://github.com/ckoven/runscripts.

**Data Availability**

FATES output files from all simulations described here are archived at http://dx.doi.org/10.15486/ngt/1569647





**Author Contribution**

CDK, RGK, RAF, JQC, BOC, MCD, JH, MH, MK, LMK, GL, EM, JFN, TP, SPS, JKS, ALSS, APW, and CX contributed to the development of FATES model. SJD,  MD, BF, NGM, HCM-L, RJN, AR, SPS, CV, APW, and SJW contributed to

observational data used for input to or testing of FATES model. CDK designed and performed model experiments and analyzed model output, with input from RGK and RAF. CDK wrote the manuscript, with input from all coauthors.

**Competing Interests**

The authors declare they have no competing interests.

**Figures**


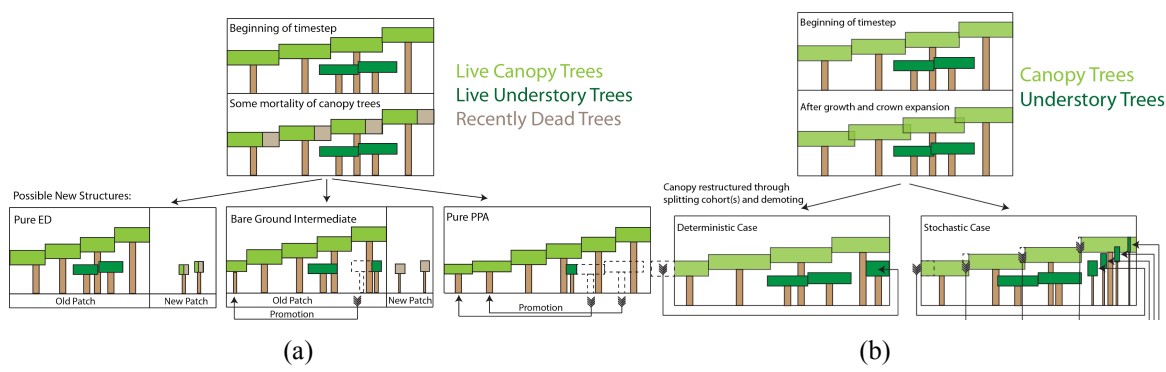

(a)                                        (b)

**Fig.1.   Schematic of how disturbance and height sorting are represented in FATES model. (a) Representation of disturbance. When canopy trees die, some fraction of the crown area of the dead trees is transferred to a newly-disturbed patch, while the remainder remains in the old patch.  Trees can be promoted from the understory to the canopy either in the old patch, or if they are transferred to the new patch area as survivors of the disturbance event. Endmembers of this case are the "pure ED" case, in which all crown area becomes new patch area, and the "pure PPA" case in which no newly disturbed patch area is created.  We**
**also consider an intermediate case, in which half of the dead tree canopy becomes disturbed, but with no survivorship of trees in the newly disturbed patch. (b) Representation of height sorting.  When canopy tree crown area exceeds the patch area that the trees are on due to crown growth, canopy trees are "demoted" to the understory. In the deterministic case, trees are rank-ordered by height and the shortest cohort is split at the point where total tree crown area equals patch area, and the remaining cohort is demoted.  In the probabilistic case, all canopy trees are demoted, with the fraction of each cohort demoted based on the cohorts'**
**relative heights.**



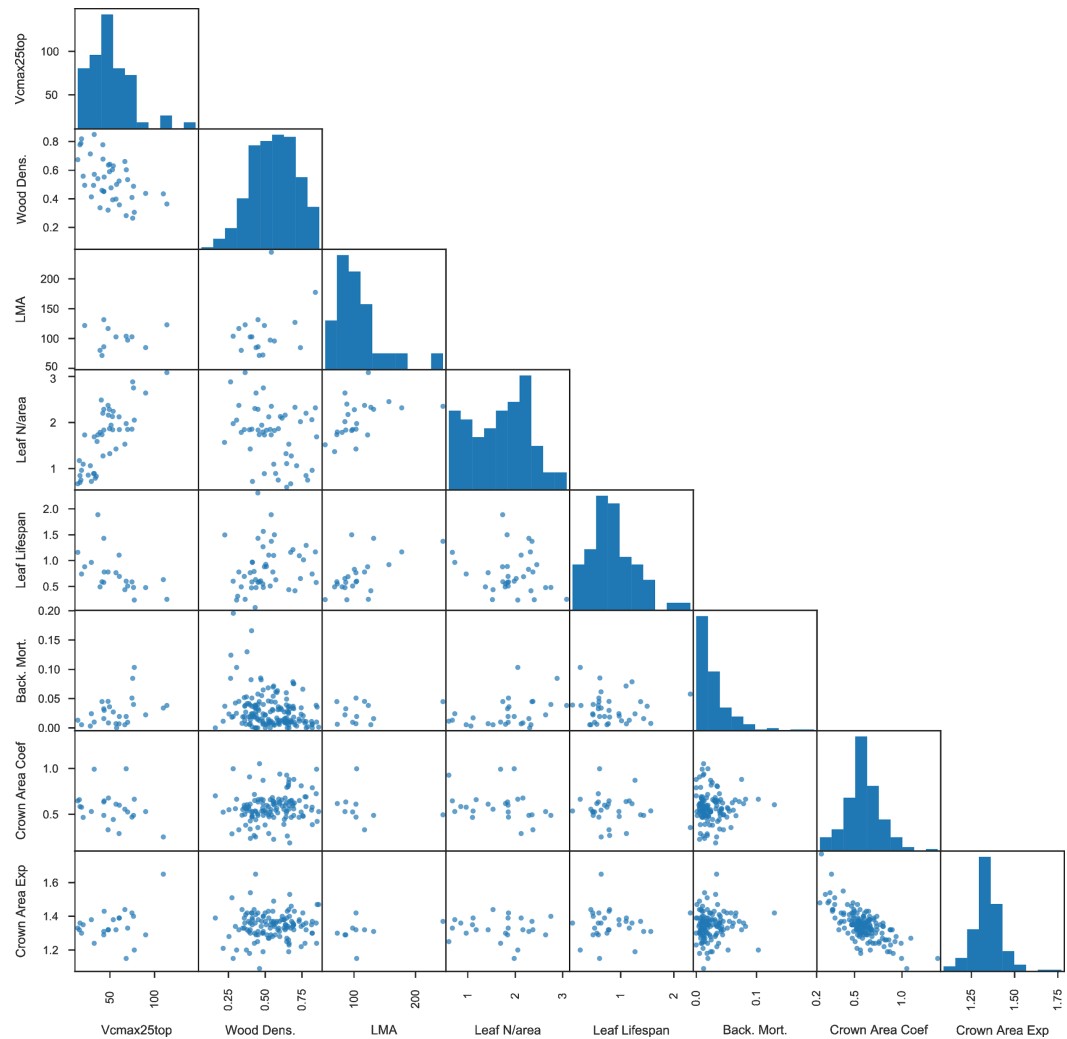

**Fig. 2 Matrix of plant trait data used to inform FATES ensembles. Measured traits are: Leaf $V_{c,max,25,top}$ (μmol $CO_2$ m$^{-2}$ s$^{-1}$), Wood Density (g/cm$^3$), Leaf Mass per unit Area (m$^2$ g$^{-1}$), Leaf N per unit Area (g m$^2$), Leaf Lifespan (y), Plant mortality rate (y$^{-1}$), Crown Area to Stem Diameter coefficient (m$^2$ cm$^{-1}$), and Crown Area to Stem Diameter exponent (unitless). Each dot represents one pair of species-level trait values where both traits are measured for a given species; histograms show the distributions of all species-level values for a given trait.**

**Biogeosciences** Open Access
Discussions

**Fig. 3 Resampled trait matrix, including 8 observed and 4 unobserved traits, as used to define PFTs in FATES simulations. 8 observed traits are as in fig 2 (with background mortality set to be the observed plant mortality). 4 Additional unobserved traits are: Allometric leaf biomass to stem diameter coefficient (kg cm$^{-1}$), allometric fine root biomass to leaf biomass ratio (unitless), fractional NPP allocated to reproduction (unitless), and maximum rate of carbon starvation mortality (yr$^{-1}$).**







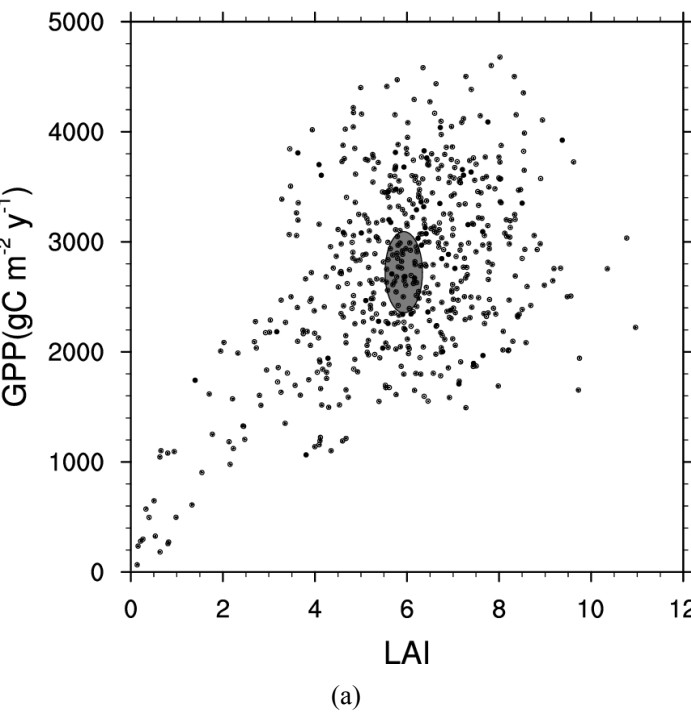

(a)

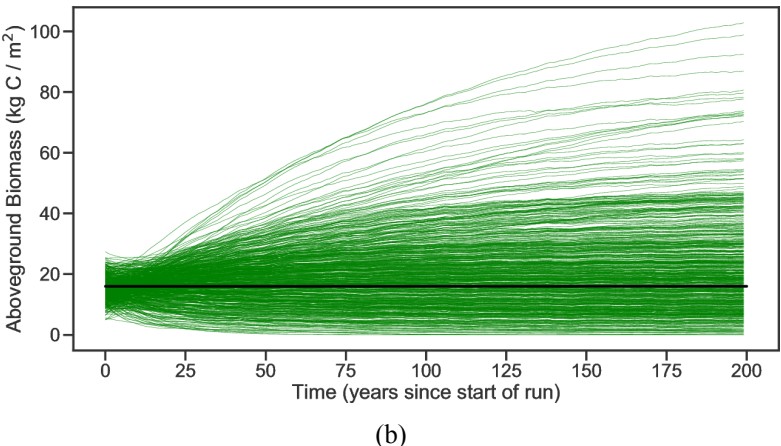

(b)


**Fig. 4 Joint distribution of modelled GPP and LAI (a) and modelled timeseries of biomass trajectories (b), for a 576-member ensemble of site-scale FATES simulations, where each ensemble member, represented here as an individual dot, has a single PFT defined as a random draw from the 12 trait covariance matrix shown in figure 2. Site-level observations of LAI and GPP (mean +/- 1 std dev) are shown in (a) as a grey ellipse, and observed mean biomass is shown in (b) as black line.**






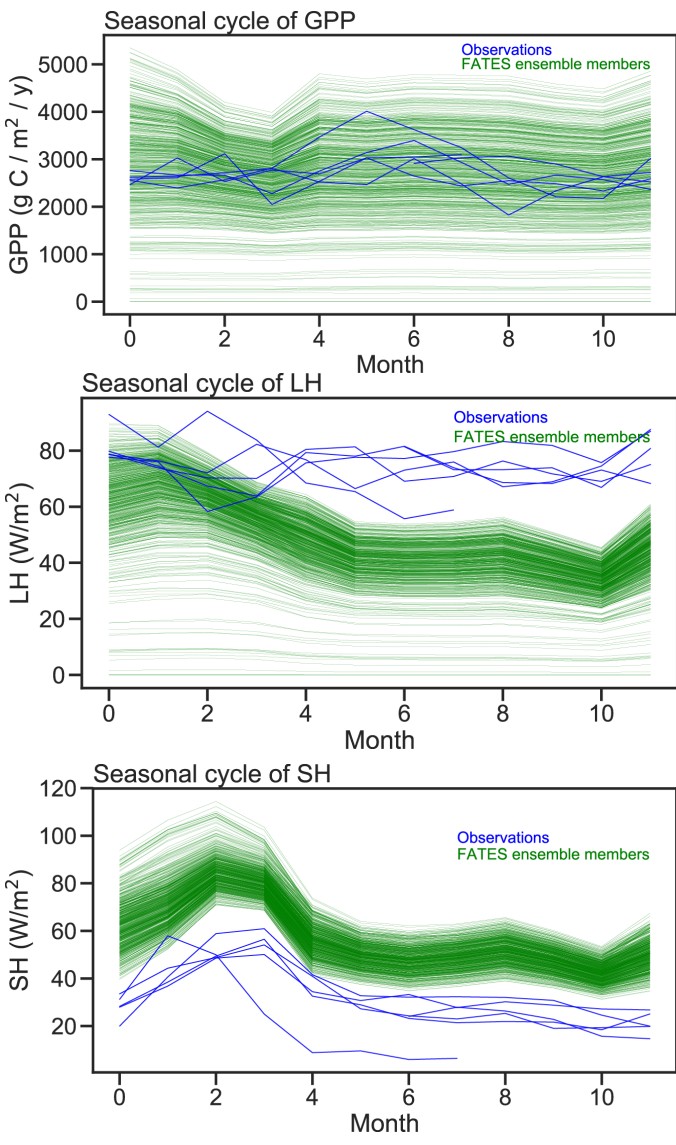


**Fig. 5 Comparison between FATES simulations of mean annual cycles in Gross Primary Productivity, Latent Heat (LH), and Sensible Heat (SH), with eddy-covariance observations from Barro Colorado Island flux tower. Green lines correspond to the mean annual cycle from each FATES ensemble member. Blue lines show individual years of eddy covariance data.**




**Fig. 6 Stem size distributions of single-PFT ensemble members, as compared to census data from the BCI forest dynamics plot**






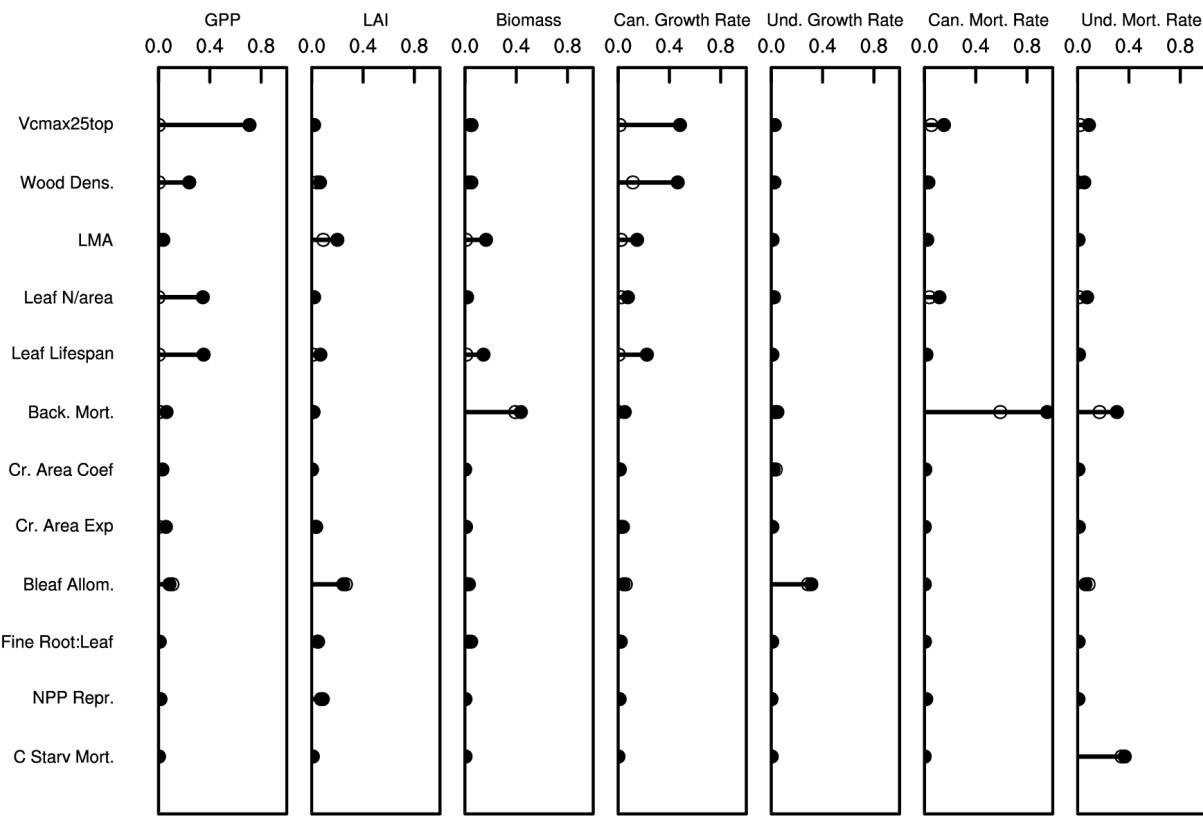

**Fig. 7 Variance decomposition of trait control on ecosystem states and vital rates. Shown are the fractions of variance explained by each of the 12 traits for 7 ecosystem variables. Filled circles and associated lines show the maximum potential fraction of variance explained by each trait, without considering trait-trait correlations. Open circles show the minimum fraction of variance explained by each trait, after first subtracting out the variance explained by all other traits.**




**Fig. 8 Comparison of FATES simulations as embedded within two land surface models: ELM-FATES and CLM-FATES. (a) GPP, (b) LAI, (c) aboveground biomass. Observational range shown as grey band in (a-c).**








**Fig. 9 Variation between ensemble members as a function of the number of competing PFTs in each ensemble member. (a-c)**
**Histograms of mean GPP (a), LAI (b), and aboveground biomass (c). Observational range shown as grey band in (a-c). (d) Size**
**distributions of ensembles. (e) Standard deviation across ensemble members as a function of the number of competing PFTs, as**
**compared to a null model, which considers the distribution changes purely as a sampling problem, for expected reduction in**
**variation between ensemble members in the absence of competition effects.**

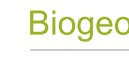
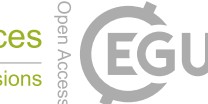







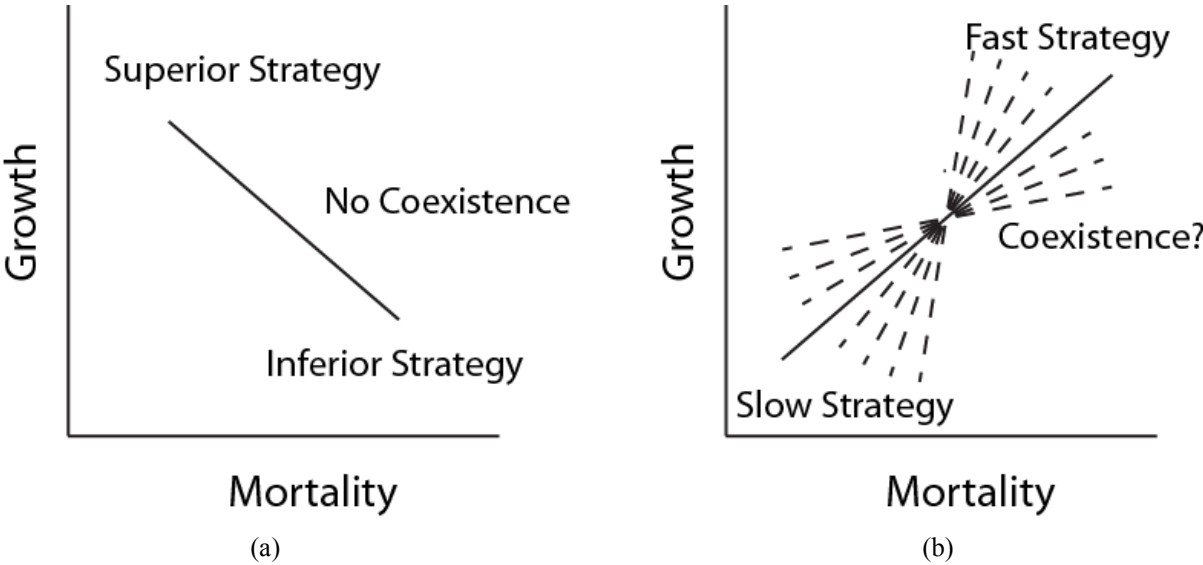

**Fig. 10 Predictions from 10-PFT ensembles with and without recruitment filtering. (a-c) Histograms of mean GPP (a), LAI (b), and aboveground biomass (c). Observational range shown as grey band in (a-c). (d) Size distributions of ensembles. (e-f) Rank abundance curves for PFTs. (e) PFT abundances for all trees greater than 1 cm diameter. (f) PFT abundances for all trees greater than 10cm diameter. For all cases in (e-f), solid line is the median ensemble member, and shaded range are the 5th-95th percentiles across 576 ensembles.**

(a)                                          (b)

**Fig. 11. Growth-mortality tradeoff and possible regimes of coexistence in a model like FATES along a successional axis. In a growth-mortality space, if a line connecting two PFTs comprising the system is negative as in (a), one PFT should be competitively dominant and exclude the other. If the slope of the line is positive as in (b) coexistence may be possible, however the range of slopes that may permit coexistence in tropical forests is not known a priori.**


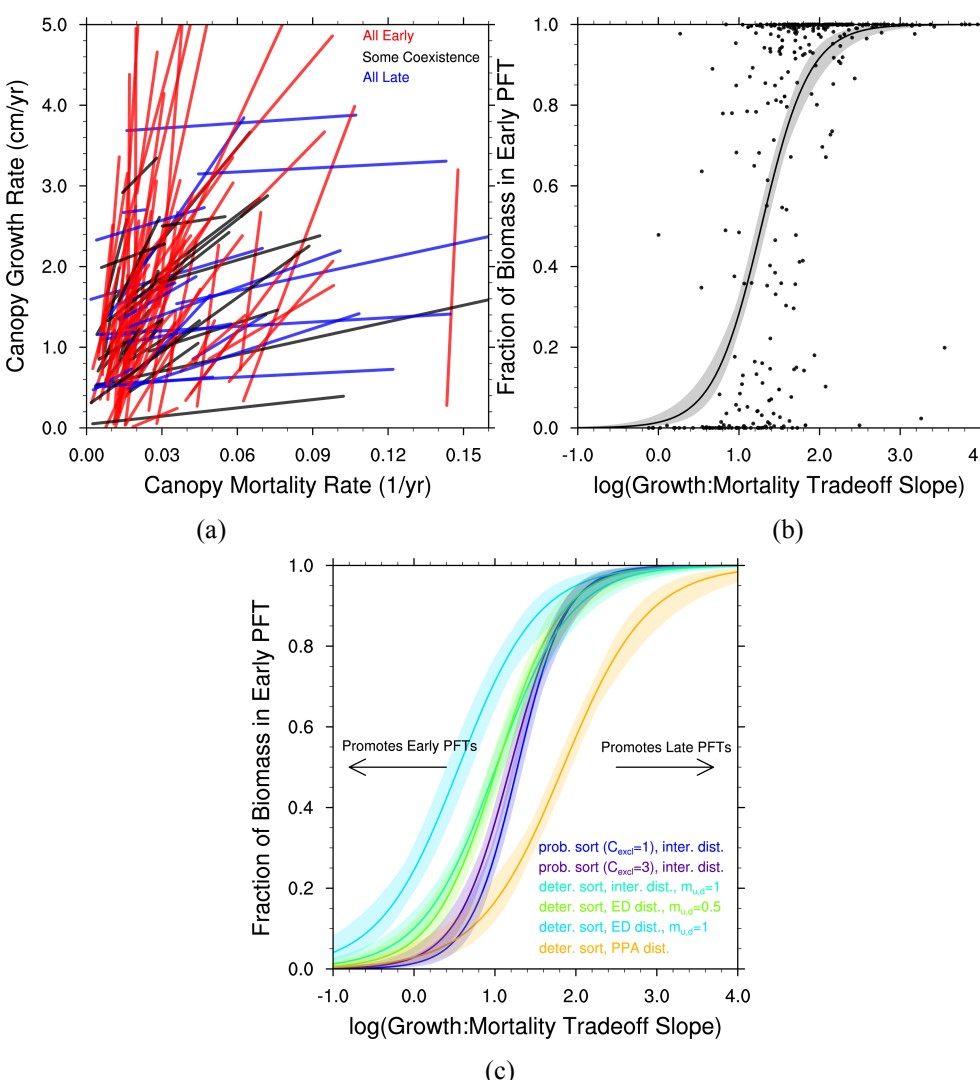


**Fig. 12 Competitive outcomes between 2 PFTs, as a function of PFT growth and mortality rates along an early-late successional continuum. Only ensemble members where a successional tradeoff, i.e. one of the PFTs both grows and dies faster than the other PFT, are shown. (a) The lines connecting paired PFTs in a growth-mortality tradeoff space, for a random subset of ensemble members in the reference case, are colored by the competitive outcome in each member to show the importance of the slope of the tradeoff line in determining the outcome. (b-c) The relative fraction of total ecosystem biomass in the faster, early successional PFT is plotted against the log of the ratios of the slope of the growth-mortality tradeoff in each PFT pairing. Curves in (b-c) show a continuous logistic regression as applied to the PFT biomass fractions in each experiment. (b) shows the individual ensemble members as well as the logistic regression for the reference case. (c) shows only the logistic regression fit for each of the cases, demonstrating that the parameter uncertainty related to disturbance and height sorting that differentiates each ensemble leads to divergent outcomes in the relative success between early and late successional PFTs. See figure 1 for qualitative schematics of the different structural cases.**





**Fig. 13. Ecosystem-level model results of paired PFT competition experiments. Ensemble distributions of (a) GPP, (b) LAI, (c) biomass, and (d) size distributions. For each of the paired-PFT cases.**






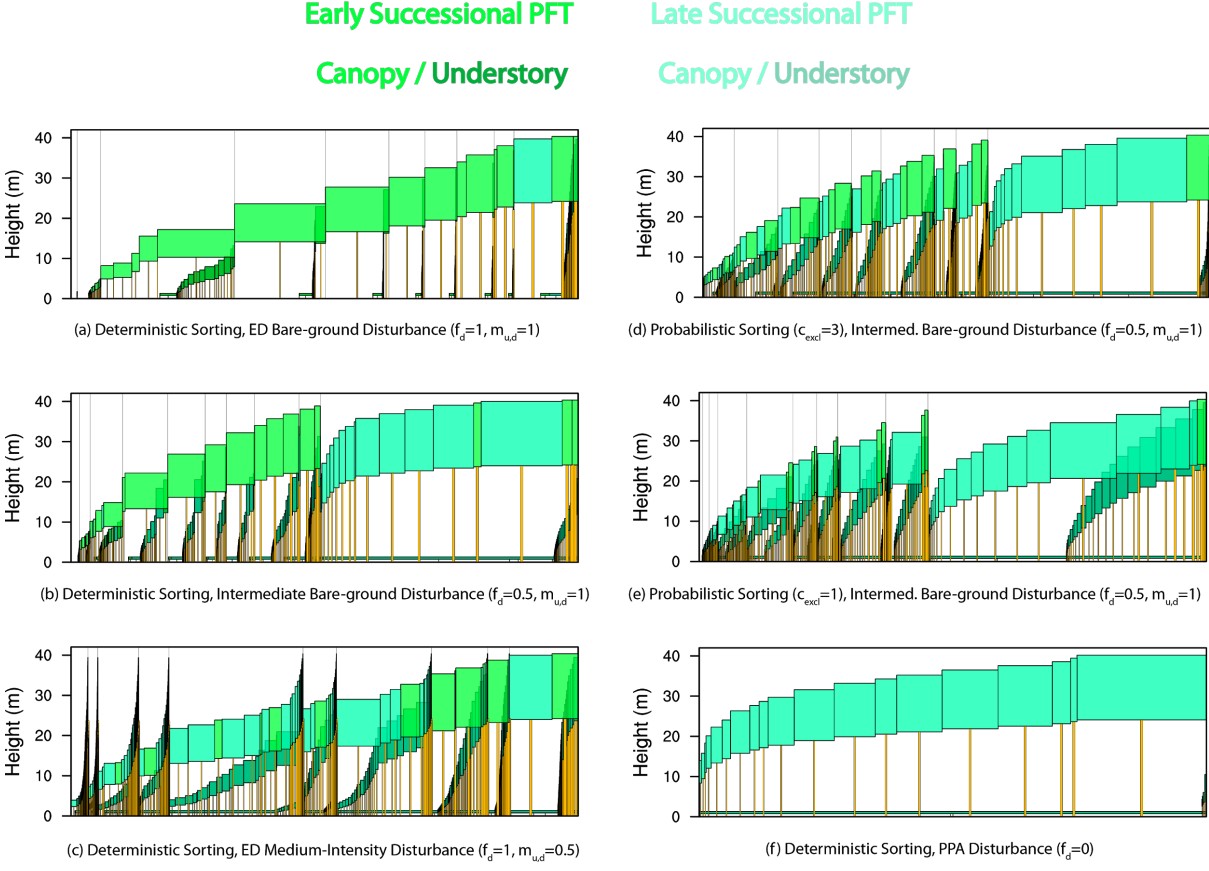

**Fig. 14. Forest structure and composition, at year 600 of one ensemble member for each structural sensitivity experiment. Experiments are as in figures 12-13, and ordered from most promoting of early successional to late successional PFTs: (a) Deterministic sorting, "Pure ED", bare ground ($f_d$ =1, $m_{u,d}$=1) disturbance; (b) Deterministic sorting, intermediate bare-ground disturbance; (c) Deterministic sorting, "Pure ED" medium intensity ($f_d$ =1, $m_{u,d}$=0.5) disturbance; (d) Probabilistic sorting ($c_{excl}$=3), intermediate bare-ground disturbance; (e) Probabilistic sorting ($c_{excl}$=1), intermediate bare-ground disturbance; and (f) Deterministic sorting, "Pure PPA" ($f_d$ =0) disturbance. The same ensemble member was used for each panel, so that plant traits are identical across experiments. Each panel depicts individual cohorts, arranged from tallest to shortest within a patch from right to left. The horizontal axis of each panel shows area: both cohort crown area and patch area. Older patches are to the right, with thin vertical lines separating patches. Cohort widths in the figure are proportional to the crown area of each cohort. Within the canopy, different PFTs are given different colors, with an early successional PFT in light green and a late successional PFT in blue-green. Understory cohorts are shaded darker than canopy cohorts.**



**Tables**

| Number of PFTs competing per ensemble member | Height Sorting | Recruitment | Disturbance | LSM | Purpose |
|---|---|---|---|---|---|
| 1 | Deterministic | normal | bare-ground intermediate: $f_d$=0.5, $m_{u,d}$=1 | CLM | Control |
| 1 | Deterministic | normal | bare-ground intermediate | ELM | Understand sensitivity to driving model |
| 3 | Deterministic | mixing | bare-ground intermediate | CLM | Understand sensitivity to number of PFTs |
| 10 | Deterministic | mixing | bare-ground intermediate | CLM | Understand sensitivity to number of PFTs |
| 10 | Deterministic | normal | bare-ground intermediate | CLM | Understand sensitivity to inter-generational trait filtering |
| 2 | Deterministic | normal | bare-ground intermediate | CLM | Reference case for looking at regimes of coexistence |
| 2 | Probabilistic, $c_{excl}$=3 | normal | bare-ground intermediate | CLM | Understand sensitivity of coexistence to representation of height sorting |
| 2 | Probabilistic, $c_{excl}$=1 | normal | bare-ground intermediate | CLM | Understand sensitivity of coexistence to representation of height sorting |
| 2 | Deterministic | normal | pure ED: $f_d$=1, $m_{u,d}$=0.5 | CLM | Understand sensitivity of coexistence to representation of disturbance |
| 2 | Deterministic | normal | pure ED: $f_d$=1, $m_{u,d}$=1.0 | CLM | Understand sensitivity of coexistence to representation of disturbance |
| 2 | Deterministic | normal | pure PPA: $f_d$=0 | CLM | Understand sensitivity of coexistence to representation of disturbance |




**Table 1. Experimental matrix used in this study. Each ensemble above consists of 576 ensemble members with one or more PFTs per ensemble member chosen as a random draw from the 12x12 trait covariance matrix.**

**Supplementary Video**

**SV1 Animated version of Fig. 13d, showing 600 years of forest development from bare ground for one ensemble member.**

**Video available at: https://doi.org/10.5446/43627**

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
