# Peer review of "Benchmarking and Parameter Sensitivity of Physiological and Vegetation Dynamics using the Functionally Assembled Terrestrial Ecosystem Simulator (FATES) at Barro Colorado Island, Panama"

_Biogeosciences, 2019_

## Referee Comment (RC1) · Anonymous Referee #1 · 4 Dec 2019

Summary: Koven and co-authors presented a model-observation comparison and parameter sensitivity analysis of a new vegetation model with explicit demography, FATES. They performed ensemble simulations in which plant traits defining plant functional types were randomly selected from observed distribution from the field site in Panama. They assessed the effect of plant traits and disturbance parameters (i.e. competition rules) on ecosystem variables. They showed that increasing the number of competing PFTs in ensemble simulations strongly shift model predictions and

that model outputs show different sensitivity to different parameters depending of that number of PFTs. They also showed that modifying competition rules shifted dominance regimes and coexistence between PFTs. They concluded that a differentiation between plant traits influenced or not by competition should be made to quantify the sources of uncertainty in vegetation demographic models.

Major comments: Despite I found this manuscript sometimes difficult to read, I really enjoyed the approach and the analysis presented in this paper. Representing plant trait variability is a very challenging, but necessary task for building the next generation of vegetation models, and this paper will be a very useful contribution for the modelling community. I only have a few comments that could help to improve the readability of the paper: - The number of traits and simulations performed in this manuscript made it difficult to follow sometimes. Some additional information could help understand the sensitivity of the model. For example, a table linking plant traits to the main processes of the model could help understand better the sensitivity results. Also, a better description of the experiments performed and why it was performed can improve the readability of the manuscript. Descriptions in Table 1 are not clear enough. - In my opinion, the comparison between ELM and CLM does not bring any interesting information to the paper if the authors don't highlight the main differences in processes. The only conclusion you can draw here is that if you use a different model, you have different results. . .. It is not informative and a bit disconnected from the trait sensitivity part. - It would also help the reader if a small discussion on the effect of allometric relationships that are described in the method section. - We have a good description of trait data used in this paper; however, we don't have any description of sites characteristics (age, diversity, disturbances, ect. . .). It is difficult for the reader to assess the performance of the results, and the comparison of results with site data is not really informative because of that. Also, is the model previously parameterized to run on that particular site? - We don't have information about the method used to randomly select traits in the 12x12 matrix. Why these particular traits? For each trait, is the selected value weighted by the distribution? Is only the range of the distribution used? What if

you make a simulation by taking the median value of each trait? How are you sure that the whole trait-space is used for the sensitivity analysis? Especially when you increase the number of PFT, you virtually increase the trait space, which can create a bias if you compare the same number of simulations (576) for 1, 3 or 10 PFTs. Also, I was wondering if it is relevant to randomly select set of traits since we know that trade-offs exist between some traits. It means that sometimes you run the model with a set of traits that has no ecological sense. In that case, is it not better just to make a classical sensitivity analysis by varying parameter values incrementally, or at least to constrain some traits based on the known trade-off? - Finally, I find that putting in parallel the benchmark and the sensitivity analysis dilute the key messages of the paper. The paper can be strengthened by mainly focusing on the sensitivity of the simulated plot characteristics to traits variability in the light of known ecological properties of the forest. For example, are the observed shifts driven by plant traits consistent with our ecological knowledge of the forest functioning? Here it would be more interesting to show that the FATES model is able to reproduce the expected ecological behavior of the forest, and if not to explain through the sensitivity analysis which parameter or process is missing or poorly represented.

I hope the authors will find my comments useful to improve their manuscript. Best,

---

## Referee Comment (RC2) · Emilie Joetzjer (Referee) · 13 Jan 2020

In this analysis, the authors explore FATES's sensibility to parameter uncertainties using mainly observation-based trait and benchmarking the outputs against BCI's forest inventory and eddy covariance data. With large ensemble simulation, using a single PFT (no competition), they first evaluate the model. FATES performs reasonably to represent physiological processes and forest structure. The authors also show the

systematic biases of the model and explain the further needed development. Adding competition lead the model to simulate higher productivity and biomass, pointing out that even tough multiple PFTs should be a more realistic configuration to represent tropical forest, the ÂńÂămono-specificÂăÂż approach performed better. They also show that, with more disturbances, the model favor early successional PFTs (over late succesional PFTs). They conclude on the importance to differentiate parameters that are or are not influenced by competition to better quantify the source of uncertainties in VDMs.

I found the paper proposed by Koven et al, fully relevant to the community. Besides, despite the complexity of the model and the exhaustive sensitivity analysis performed, the paper is relatively easy to follow, the model description is crystal clear and the key messages are well highlighted. I particularly appreciated the effort of the authors to explain the sensitivity analysis through the ecological processes undergoing in the model. Therefore I think the paper is ready for publication, and I only have minors comments (or questions) detailed below.

**Comments**

In the introduction, it might be nice for non-landsurfacemodelling reader to sum up in a sentence the connections between ESM, LSM and VDM.

L163: Is RH calculated by the LSM or by FATES?

L 282: Units?

L315: Can you describe a bit more BCI and BCI data? Also, what are the time period of the eddy fluxes data? Did you recycled the meteorological data from 1986-2017 to force CLM-FATES several hundreds years, or did you took an averaged climate?

L369 I didn't get why each simulation is initialized with the observed size distribution. I would guess that the size distribution of a population will strongly depends on the PFT definition. Because FATES is able to compute the size distribution, why not start the

simulation from scratch?

L434 Might be a naive question, but do you cover a large enough range of the observed variability by taking the same number of ensemble for 1PFT, 3PFT and 10PFT?

L530 While I think it's interesting to show how insensitive is FATE to the Âńâǎhost modelÂǎÂż by comparing CLM-FATE and ELM-FATE, I wonder how relevant this analysis is. Indeed, both model are forced by the same atmospheric conditions, and as pointed by the author, the only difference reside in the soil representation. My guess, is that both host models provide enough soil moisture to FATE (because they are likely to share the same soil parameterization and BCI is relatively wet), therefore it is logical that FATES behave similarly. I wonder what is the point of this test.

L615 I'm wondering how climate variability can play a role in simulating species co-existence (thus my question on the forcing file).

L758 While I agree with your conclusion, you might want to be a bit more explicit. VDM are certainly one of the way to go in ESM to better quantify how environmental changes will affect ecosystems and the associated feedbacks. In the analysis, you show how large (and unconstrained) can be the effect of competition, and how difficult simulating co-existence is. However, the comparison with the observed data suggest that 1 PFT perform usually better than the simulation that integrate competition (Fig. 9). I think it would be nice to have a sentence bridging complexity (and reliability) vs. simple (and therefore easy-to-tune) models.

Fig. 6 You might want to give more detail on the units of "Tree number density". Is it a number of tree per ha, per diameter class. Idem on Fig. 13.

Fig. 9e Is the log scale necessary?

---

## Author Response (AR1)

**Author responses to Editor**
(Summary of revisions in green)

Dear authors,
many thanks for your responses. Please submit a suitably revised manuscript along with a point-by-point reply as to how you have addressed the comments in the revisions.
Best wishes,
Sönke Zaehle

Dear Dr Zaehle,
Thank you for your careful review of the documents. Please find the revised manuscript and point-by-point responses to the reviewers.
Yours,
Charlie Koven

**Author responses to Anonymous Referee #1**
**(Responses in Blue)** (Summary of revisions in green)

Summary: Koven and co-authors presented a model-observation comparison and parameter sensitivity analysis of a new vegetation model with explicit demography, FATES. They performed ensemble simulations in which plant traits defining plant functional types were randomly selected from observed distribution from the field site in Panama. They assessed the effect of plant traits and disturbance parameters (i.e. competition rules) on ecosystem variables. They showed that increasing the number of competing PFTs in ensemble simulations strongly shift model predictions and that model outputs show different sensitivity to different parameters depending of that number of PFTs. They also showed that modifying competition rules shifted dominance regimes and coexistence between PFTs. They concluded that a differentiation between plant traits influenced or not by competition should be made to quantify the sources of uncertainty in vegetation demographic models.

Major comments: Despite I found this manuscript sometimes difficult to read, I really enjoyed the approach and the analysis presented in this paper. Representing plant trait variability is a very challenging, but necessary task for building the next generation of vegetation models, and this paper will be a very useful contribution for the modelling community. I only have a few comments that could help to improve the readability of the paper:

We thank the reviewer for their careful review and positive assessment of the manuscript.

- The number of traits and simulations performed in this manuscript made it difficult to follow sometimes. Some additional information could help understand the sensitivity of the model. For example, a table linking plant traits to the main processes of the model could help understand better the sensitivity results. Also, a better description of the experiments performed and why it was performed can improve the readability of the manuscript. Descriptions in Table 1 are not clear enough.

In the revised manuscript we will either either a more information on the meaning of the parameters and why they were chosen into table 1; and/or include a box-and-wire process diagram, with some details of how some of the traits used here fit in to different aspects of the model, so as to provide an overview of why we selected these traits.

We have added an additional figure and table to clarify these points. The new figure 1 includes a diagram of processes in the model, which are then referred to in the new table 1, which lists each trait that we varied and which process it is most closely associated with.

- In my opinion, the comparison between ELM and CLM does not bring any interesting information to the paper if the authors don't highlight the main differences in processes. The only conclusion you can draw here is that if you use a different model, you have different results. . .. It is not informative and a bit disconnected from the trait sensitivity part.

We agree with both reviewers that the justification for this analysis was not sufficiently made in the submitted draft. We do feel that this is important to include in the manuscript, because this approach of modularizing a distinct component of the land surface into a separate codebase that can be called by multiple LSMs (and ESMs) represents an important strategy in managing process complexity and shifting away from a model-centered approach and towards a hypothesis-centered approach to experimental design in simulating the land surface. It is thus important to document the dynamics of FATES within each of the LSMs it currently works within in order to test the hypothesis that such an approach is valid. We will add further discussion of this point in the revisions, as well as linking to a separate manuscript by authors RAF and CDK, which explores this point in further detail and is currently in review.

We have added the following in the revisions: "This modularity of FATES and its ability to work within alternate LSMs represents an important capability. As LSMs have grown ever more complex, the infrastructure for managing model complexity and attributing model behavioral differences to structural and parametric assumptions has not grown equivalently; a potential strategy for addressing this complexity problem is to separate the representation of processes in such a way that they can be explored as conditional on alternate boundary conditions, following the 'modular complexity' approach described in Fisher and Koven, (*in review*). Such an approach can allow, e.g., one to ask how the representation of soil physics of biogeochemistry feeds back onto an identical representation of plant physiology in order to better separate the contributions of each to total model prediction uncertainty. Here we begin to test this approach, by testing the exact same representation of FATES within the alternate LSMs it can be run in."

- It would also help the reader if a small discussion on the effect of allometric relationships that are described in the method section.

We still haven't fully explored these differences in the model, and leave this as future work. We will discuss this point in the revised manuscript.

We have added the sentence: "We have not yet fully explored the sensitivity of model dynamics to the alternate aboveground biomass allometries; this remains future work."

- We have a good description of trait data used in this paper; however, we don't have any description of sites characteristics (age, diversity, disturbances, ect. . .). It is difficult for the reader to assess the performance of the results, and the comparison of results with site data is not really informative because of that. Also, is the model previously parameterized to run on that particular site?

We'll add more info on the site in the revised version of the manuscript.

We have added a paragraph describing the site in more detail in the revised:
"All model experiments here are conducted at Barro Colorado Island (BCI), Panama (9.151°N, 79.855°W). The environment at BCI has a mean precipitation of 2600 +- 480 mm/yr, with a 4-month dry season during which precipitation drops below 100 mm/yr.  The ecosystem at BCI is a primary forest, with a disturbance regime characterized by primarily small-scale disturbance and subject to elevated mortality rates during period ENSO-driven droughts.  The site includes a 50 ha census plot, in which every stem ≥1cm diameter has been measured every 5 years since 1982, with 321 species identified (Condit et al., 2017), as well as eddy covariance and other observations."

- We don't have information about the method used to randomly select traits in the 12x12 matrix. Why these particular traits? For each trait, is the selected value weighted by the distribution? Is only the range of the distribution used? What if you make a simulation by taking the median value of each trait? How are you sure that the whole trait-space is used for the sensitivity analysis? Especially when you increase the number of PFT, you virtually increase the trait space, which can create a bias if you compare the same number of simulations (576) for 1, 3 or 10 PFTs. Also, I was wondering if it is relevant to randomly select set of traits since we know that trade-offs exist between some traits. It means that sometimes you run the model with a set of traits that has no ecological sense. In that case, is it not better just to make a classical sensitivity analysis by varying parameter values incrementally, or at least to constrain some traits based on the known trade-off?

We randomly sample the traits, assuming either normal or lognormal distributions for all traits, and allow any covariances that exist in the data to be propagated into the trait sampling.  Because of the nonzero covariances and (for some traits at least) known trait distributions, we chose the method that we use here.

- Finally, I find that putting in parallel the benchmark and the sensitivity analysis dilute the key messages of the paper. The paper can be strengthened by mainly focusing on the sensitivity of the simulated plot characteristics to traits variability in the light of known ecological properties of the forest. For example, are the observed shifts driven by plant traits consistent with our ecological knowledge of the forest functioning? Here it would be more interesting to show that the FATES model is able to reproduce the expected ecological behavior of the forest, and if not to explain through the sensitivity analysis which parameter or process is missing or poorly represented.

We want to combine benchmarking and parameter sensitivity together in order to (a) establish the usefulness of the model for describing the ecosystem, and (b) because the

agreement of model predictions to observations are important in understanding some of the sensitivity questions that we explore in the manuscript, such as in sections 3.3 and 3.4.

I hope the authors will find my comments useful to improve their manuscript. Best, Interactive comment on Biogeosciences Discuss., https://doi.org/10.5194/bg-2019-409, 2019.

We thank the reviewer for their comments.

**Author Responses to Emilie Joetzjer (Referee)**  **(Responses in Blue)**

In this analysis, the authors explore FATES's sensibility to parameter uncertainties using mainly observation-based trait and benchmarking the outputs against BCI's forest inventory and eddy covariance data. With large ensemble simulation, using a single PFT (no competition), they first evaluate the model. FATES performs reasonably to represent physiological processes and forest structure. The authors also show the systematic biases of the model and explain the further needed development. Adding competition leads the model to simulate higher productivity and biomass, pointing out that even though multiple PFTs should be a more realistic configuration to represent tropical forest, the "mono-specific" approach performed better. They also show that, with more disturbances, the model favor early successional PFTs (over late successional PFTs). They conclude on the importance to differentiate parameters that are or are not influenced by competition to better quantify the source of uncertainties in VDMs.

I found the paper proposed by Koven et al, fully relevant to the community. Besides, despite the complexity of the model and the exhaustive sensitivity analysis performed, the paper is relatively easy to follow, the model description is crystal clear and the key messages are well highlighted. I particularly appreciated the effort of the authors to explain the sensitivity analysis through the ecological processes undergoing in the model. Therefore I think the paper is ready for publication, and I only have minors comments (or questions) detailed below.

We thank Dr Joetzjer for her careful review and positive assessment of the manuscript.

**Comments**

In the introduction, it might be nice for non-land surface modelling reader to sum up in a sentence the connections between ESM, LSM and VDM.

We will clarify these distinctions in the revised text.

In the revisions, we have added the following sentence. "A VDM is a size- and age-structured representation of vegetation dynamics within an LSM, and which may also be coupled within an ESM."

L163: Is RH calculated by the LSM or by FATES?

As the domain of FATES is only the vegetative part of the ecosystem, we allow RH to be external to FATES and thus calculated by the LSM.

In the revisions, we have added the sentence: "Heterotrophic respiration is handled outside of FATES, by the LSM it is embedded within."

L 282: Units?

$M_c$ is the rate of mortality of crown area of plants in the canopy strata, and is tracked in $m^2$/ha/year. $F_d$ is unitless, and thus $R_d$ is also in $m^2$/ha/year. We will clarify this in the revised version.

We have added units here.

L315: Can you describe a bit more BCI and BCI data? Also, what are the time period of the eddy fluxes data? Did you recycled the meteorological data from 1986-2017 to force CLM-FATES several hundreds years, or did you took an averaged climate?

We will describe the site more in the revised version. We recycled the meteorological data over the course of the model simulations.

We have added a paragraph describing the site in more detail in the revised:
"All model experiments here are conducted at Barro Colorado Island (BCI), Panama (9.151°N, 79.855°W). The environment at BCI has a mean precipitation of 2600 +- 480 mm/yr, with a 4-month dry season during which precipitation drops below 100 mm/yr. The ecosystem at BCI is a primary forest, with a disturbance regime characterized by primarily small-scale disturbance and subject to elevated mortality rates during period ENSO-driven droughts. The site includes a 50 ha census plot, in which every stem ≥1cm diameter has been measured every 5 years since 1982, with 321 species identified (Condit et al., 2017), as well as eddy covariance and other observations."

We also added the sentence "For each simulation, we re-cycled meteorology over the 1986-2017 period."

L369 I didn't get why each simulation is initialized with the observed size distribution. I would guess that the size distribution of a population will strongly depends on the PFT definition. Because FATES is able to compute the size distribution, why not start the simulation from scratch?

The initialization of the size distribution is only to accelerate the convergence of the model. We have found that when we initialize the model from bare ground, we arrive at similar size distributions. Because of the large number of ensembles and experiments here, we chose to initialize the size distributions from census rather than bare ground so as to reduce computational costs.

L434 Might be a naive question, but do you cover a large enough range of the observed variability by taking the same number of ensemble for 1PFT, 3PFT and 10PFT?

This is a good question, and why we chose a relatively large number of ensemble members in each of the simulations: by starting with a relatively deep sampling of the trait space in the 1PFT simulations, we don't expect a qualitatively different behavior due only to that when using larger a number of PFTs. We do sample more parts of the trait space when using a larger number of PFTs and the same number of ensemble members; however, we note that the effect of this larger sampling is actually to decrease the width of the distributions across ensemble members, due to the competitive dynamics described in the manuscript. Thus the

more thorough sampling of the trait space itself is unlikely to contribute strongly to that signal.

L530 While I think it's interesting to show how insensitive is FATE to the "host model" by comparing CLM-FATE and ELM-FATE, I wonder how relevant this analysis is. Indeed, both model are forced by the same atmospheric conditions, and as pointed by the author, the only difference reside in the soil representation. My guess, is that both host models provide enough soil moisture to FATE (because they are likely to share the same soil parameterization and BCI is relatively wet), therefore it is logical that FATES behave similarly. I wonder what is the point of this test.

We agree with both reviewers that the justification for this analysis was not sufficiently made in the submitted draft. We do feel that this is important to include in the manuscript, because this approach of modularizing a distinct component of the land surface into a separate codebase that can be called by multiple LSMs (and ESMs) represents an important strategy in managing process complexity and shifting away from a model-centered approach and towards a hypothesis-centered approach to experimental design in simulating the land surface. It is thus important to document the dynamics of FATES within each of the LSMs it currently works within in order to test the hypothesis that such an approach is valid. We will add further discussion of this point in the revisions, as well as linking to a separate manuscript by authors RAF and CDK, which explores this point in further detail and is currently in review.

We have added the following in the revisions: "This modularity of FATES and its ability to work within alternate LSMs represents an important capability. As LSMs have grown ever more complex, the infrastructure for managing model complexity and attributing model behavioral differences to structural and parametric assumptions has not grown equivalently; a potential strategy for addressing this complexity problem is to separate the representation of processes in such a way that they can be explored as conditional on alternate boundary conditions, following the 'modular complexity' approach described in Fisher and Koven, (*in review*). Such an approach can allow, e.g., one to ask how the representation of soil physics of biogeochemistry feeds back onto an identical representation of plant physiology in order to better separate the contributions of each to total model prediction uncertainty. Here we begin to test this approach, by testing the exact same representation of FATES within the alternate LSMs it can be run in."

L615 I'm wondering how climate variability can play a role in simulating species coexistence (thus my question on the forcing file).

This is a great question, which we don't get into here. This question is explored more fully using the ED2-hydro model and the same meteorological dataset in Powell et al., (2018, https://doi.org/10.1111/nph.15271).

L758 While I agree with your conclusion, you might want to be a bit more explicit. VDM are certainly one of the way to go in ESM to better quantify how environmental changes will affect ecosystems and the associated feedbacks. In the analysis, you show how large (and unconstrained) can be the effect of competition, and how difficult simulating co-existence is.

However, the comparison with the observed data suggest that 1 PFT perform usually better than the simulation that integrate competition (Fig. 9). I think it would be nice to have a sentence bridging complexity (and reliability) vs. simple (and therefore easy-to-tune) models.

In the revised manuscript, we will explore in greater detail these caveats and difficulties associated with using this type of model structure, and the costs and benefits of choosing different positions along this axis of model complexity.

We have rephrased this as the following: "VDMs are a promising tool to resolve these processes, however VDMs bring a high degree of complexity that add greater uncertainty to model predictions than more simple model frameworks that may be more easily tuned to match observations."

Fig. 6 You might want to give more detail on the units of "Tree number density". Is it a number of tree per ha, per diameter class. Idem on Fig. 13.

These units are in n/ha/cm, with the $cm^{-1}$ referring to the width of the size bins used for plotting the curve. Will add this to the figure description in the revised manuscript.

We have added this description to the legend of figure 6.

Fig. 9e Is the log scale necessary?

The choice we faced was either to plot the log of the slope or the angle of the slope, as we didn't want to make the sensitivity asymmetric between high-slope and low-slope parts of the domain. We chose a log scale here, we will explore in revisions whether there is a better approach to showing this relationship.

We have decided to keep the log scale here.

[revised manuscript text omitted]

---

## Author Response (AR2)

**Author responses to Editor**

(Summary of revisions in green)

Comments to the Author:
Dear authors,

many thanks for the revised manuscript which I find suitable for publication in Biogeosciences subject to the following technical edit: Unless the paper is now accepted for publication, please remove the reference to Fisher & Koven in review, i.e. the subclause ", following the 'modular complexity' approach described in Fisher and Koven, ( in review )".

Best wishes,
Sönke Zaehle

Dear Dr Zaehle,

Thank you again for your careful review of the documents. The Fisher and Koven manuscript has now been accepted by *JAMES*, and assigned a provisional DOI, and we have updated the text and references accordingly. In addition, we have gone through the text one more time to fix typos and consolidate all multi-panel figures into single files.

Yours,
Charlie Koven

[revised manuscript text omitted]